# Improving Regret Bounds for Combinatorial Semi-Bandits with Probabilistically Triggered Arms and Its Applications

**Qinshi Wang**
Princeton University
Princeton, NJ 08544
qinshiw@princeton.edu

**Wei Chen**
Microsoft Research
Beijing, China
weic@microsoft.com

## Abstract

We study combinatorial multi-armed bandit with probabilistically triggered arms and semi-bandit feedback (CMAB-T). We resolve a serious issue in the prior CMAB-T studies where the regret bounds contain a possibly exponentially large factor of $1/p^*$, where $p^*$ is the minimum positive probability that an arm is triggered by any action. We address this issue by introducing a triggering probability modulated (TPM) bounded smoothness condition into the general CMAB-T framework, and show that many applications such as influence maximization bandit and combinatorial cascading bandit satisfy this TPM condition. As a result, we completely remove the factor of $1/p^*$ from the regret bounds, achieving significantly better regret bounds for influence maximization and cascading bandits than before. Finally, we provide lower bound results showing that the factor $1/p^*$ is unavoidable for general CMAB-T problems, suggesting that the TPM condition is crucial in removing this factor.

## 1 Introduction

Stochastic multi-armed bandit (MAB) is a classical online learning framework modeled as a game between a player and the environment with $m$ arms. In each round, the player selects one arm and the environment generates a reward of the arm from a distribution unknown to the player. The player observes the reward, and use it as the feedback to the player's algorithm (or policy) to select arms in future rounds. The goal of the player is to cumulate as much reward as possible over time. MAB models the classical dilemma between exploration and exploitation: whether the player should keep exploring arms in search for a better arm, or should stick to the best arm observed so far to collect rewards. The standard performance measure of the player's algorithm is the *(expected) regret*, which is the difference in expected cumulative reward between always playing the best arm in expectation and playing according to the player's algorithm.

In recent years, stochastic combinatorial multi-armed bandit (CMAB) receives many attention (e.g. [9, 7, 6, 10, 13, 15, 14, 16, 8]), because it has wide applications in wireless networking, online advertising and recommendation, viral marketing in social networks, etc. In the typical setting of CMAB, the player selects a combinatorial action to play in each round, which would trigger the play of a set of arms, and the outcomes of these triggered arms are observed as the feedback (called semi-bandit feedback). Besides the exploration and exploitation tradeoff, CMAB also needs to deal with the exponential explosion of the possible actions that makes exploring all actions infeasible.

One class of the above CMAB problems involves probabilistically triggered arms [7, 14, 16], in which actions may trigger arms probabilistically. We denote it as CMAB-T in this paper. Chen et al. [7] provide such a general model and apply it to the influence maximization bandit, which models

stochastic influence diffusion in social networks and sequentially selecting seed sets to maximize the cumulative influence spread over time. Kveton et al. [14, 16] study cascading bandits, in which arms are probabilistically triggered following a sequential order selected by the player as the action. However, in both studies, the regret bounds contain an undesirable factor of $1/p^*$, where $p^*$ is the minimum positive probability that any arm can be triggered by any action,[1] and this factor could be exponentially large for both influence maximization and cascading bandits.

In this paper, we adapt the general CMAB framework of [7] in a systematic way to completely remove the factor of $1/p^*$ for a large class of CMAB-T problems including both influence maximization and combinatorial cascading bandits. The key observation is that for these problems, a harder-to-trigger arm has less impact to the expected reward and thus we do not need to observe it as often. We turn this key observation into a triggering probability modulated (TPM) bounded smoothness condition, adapted from the original bounded smoothness condition in [7]. We eliminates the $1/p^*$ factor in the regret bounds for all CMAB-T problems with the TPM condition, and show that influence maximization bandit and the conjunctive/disjunctive cascading bandits all satisfy the TPM condition. Moreover, for general CMAB-T without the TPM condition, we show a lower bound result that $1/p^*$ is unavoidable, because the hard-to-trigger arms are crucial in determining the best arm and have to be observed enough times.

Besides removing the exponential factor, our analysis is also tighter in other regret factors or constants comparing to the existing influence maximization bandit results [7, 25], combinatorial cascading bandit [16], and linear bandits without probabilistically triggered arms [15]. Both the regret analysis based on the TPM condition and the proof that influence maximization bandit satisfies the TPM condition are technically involved and nontrivial, but due to the space constraint, we have to move the complete proofs to the supplementary material. Instead we introduce the key techniques used in the main text.

**Related Work.** Multi-armed bandit problem is originally formated by Robbins [20], and has been extensively studied in the literature [cf. 3, 21, 4]. Our study belongs to the stochastic bandit research, while there is another line of research on adversarial bandits [2], for which we refer to a survey like [4] for further information. For stochastic MABs, an important approach is Upper Confidence Bound (UCB) approach [1], on which most CMAB studies are based upon.

As already mentioned in the introduction, stochastic CMAB has received many attention in recent years. Among the studies, we improve (a) the general framework with probabilistically triggered arms of [7], (b) the influence maximization bandit results in [7] and [25], (c) the combinatorial cascading bandit results in [16], and (d) the linear bandit results in [15]. We defer the technical comparison with these studies to Section 4.3. Other CMAB studies do not deal with probabilistically triggered arms. Among them, [9] is the first study on linear stochastic bandit, but its regret bound has since been improved by Chen et al. [7], Kveton et al. [15]. Combes et al. [8] improve the regret bound of [15] for linear bandits in a special case where arms are mutually independent. Most studies above are based on the UCB-style CUCB algorithm or its minor variant, and differ on the assumptions and regret analysis. Gopalan et al. [10] study Thompson sampling for complex actions, which is based on the Thompson sample approach [22] and can be applied to CMAB, but their regret bound has a large exponential constant term.

Influence maximization is first formulated as a discrete optimization problem by Kempe et al. [12], and has been extensively studied since (cf. [5]). Variants of influence maximization bandit have also been studied [18, 23, 24]. Lei et al. [18] use a different objective of maximizing the expected size of the union of the influenced nodes over time. Vaswani et al. [23] discuss how to transfer node level feedback to the edge level feedback, and then apply the result of [7]. Vaswani et al. [24] replace the original maximization objective of influence spread with a heuristic surrogate function, avoiding the issue of probabilistically triggered arms. But their regret is defined against a weaker benchmark relaxed by the approximation ratio of the surrogate function, and thus their theoretical result is weaker than ours.

## 2 General Framework

In this section we present the general framework of combinatorial multi-armed bandit with probabilistically triggered arms originally proposed in [7] with a slight adaptation, and denote it as CMAB-T. We illustrate that the influence maximization bandit [7] and combinatorial cascading bandits [14, 16] are example instances of CMAB-T.

CMAB-T is described as a learning game between a learning agent (or player) and the environment. The environment consists of $m$ random variables $X_1, \ldots, X_m$ called *base arms* (or *arms*) following a joint distribution $D$ over $[0,1]^m$. Distribution $D$ is picked by the environment from a class of distributions $\mathcal{D}$ before the game starts. The player knows $\mathcal{D}$ but not the actual distribution $D$.

The learning process proceeds in discrete rounds. In round $t \geq 1$, the player selects an action $S_t$ from an action space $\mathcal{S}$ based on the feedback history from the previous rounds, and the environment draws from the joint distribution $D$ an independent sample $X^{(t)} = (X_1^{(t)}, \ldots, X_m^{(t)})$. When action $S_t$ is played on the environment outcome $X^{(t)}$, a random subset of arms $\tau_t \subseteq [m]$ are triggered, and the outcomes of $X_i^{(t)}$ for all $i \in \tau_t$ are observed as the feedback to the player. The player also obtains a nonnegative reward $R(S_t, X^{(t)}, \tau_t)$ fully determined by $S_t, X^{(t)}$, and $\tau_t$. A learning algorithm aims at properly selecting actions $S_t$'s over time based on the past feedback to cumulate as much reward as possible. Different from [7], we allow the action space $\mathcal{S}$ to be infinite. In the supplementary material, we discuss an example of continuous influence maximization [26] that uses continuous and infinite action space while the number of base arms is still finite.

We now describe the triggered set $\tau_t$ in more detail, which is not explicit in [7]. In general, $\tau_t$ may have additional randomness beyond the randomness of $X^{(t)}$. Let $D^{\text{trig}}(S, X)$ denote a distribution of the triggered subset of $[m]$ for a given action $S$ and an environment outcome $X$. We assume that $\tau_t$ is drawn independently from $D^{\text{trig}}(S_t, X^{(t)})$. We refer $D^{\text{trig}}$ as the *probabilistic triggering function*.

To summarize, a *CMAB-T problem instance* is a tuple $([m], \mathcal{S}, \mathcal{D}, D^{\text{trig}}, R)$, with elements already described above. These elements are known to the player, and hence establishing the problem input to the player. In contrast, the *environment instance* is the actual distribution $D \in \mathcal{D}$ picked by the environment, and is unknown to the player. The problem instance and the environment instance together form the *(learning) game instance*, in which the learning process would unfold. In this paper, we fix the environment instance $D$, unless we need to refer to more than one environment instances.

For each arm $i$, let $\mu_i = \mathbb{E}_{X \sim D}[X_i]$. Let vector $\boldsymbol{\mu} = (\mu_1, \ldots, \mu_m)$ denote the expectation vector of arms. Note that vector $\boldsymbol{\mu}$ is determined by $D$. Same as in [7], we assume that the expected reward $\mathbb{E}[R(S, X, \tau)]$, where the expectation is taken over $X \sim D$ and $\tau \sim D^{\text{trig}}(S, X)$, is a function of action $S$ and the expectation vector $\boldsymbol{\mu}$ of the arms. Henceforth, we denote $r_S(\boldsymbol{\mu}) \triangleq \mathbb{E}[R(S, X, \tau)]$. We remark that Chen et al. [6] relax the above assumption and consider the case where the entire distribution $D$, not just the mean of $D$, is needed to determine the expected reward. However, they need to assume that arm outcomes are mutually independent, and they do not consider probabilistically triggered arms. It might be interesting to incorporate probabilistically triggered arms into their setting, but this is out of the scope of the current paper. To allow algorithm to estimate $\mu_i$ directly from samples, we assume the outcome of an arm does not depend on whether itself is triggered, i.e. $\mathbb{E}_{X \sim D, \tau \sim D^{\text{trig}}(S,X)}[X_i \mid i \in \tau] = \mathbb{E}_{X \sim D}[X_i]$.

The performance of a learning algorithm $A$ is measured by its *(expected) regret*, which is the difference in expected cumulative reward between always playing the best action and playing actions selected by algorithm $A$. Formally, let $\text{opt}_{\boldsymbol{\mu}} = \sup_{S \in \mathcal{S}} r_S(\boldsymbol{\mu})$, where $\boldsymbol{\mu} = \mathbb{E}_{X \sim D}[X]$, and we assume that $\text{opt}_{\boldsymbol{\mu}}$ is finite. Same as in [7], we assume that the learning algorithm has access to an offline $(\alpha, \beta)$-*approximation oracle* $\mathcal{O}$, which takes $\boldsymbol{\mu} = (\mu_1, \ldots, \mu_m)$ as input and outputs an action $S^{\mathcal{O}}$ such that $\Pr\{r_{\boldsymbol{\mu}}(S^{\mathcal{O}}) \geq \alpha \cdot \text{opt}_{\boldsymbol{\mu}}\} \geq \beta$, where $\alpha$ is the *approximation ratio* and $\beta$ is the success probability. Under the $(\alpha, \beta)$-approximation oracle, the benchmark cumulative reward should be the $\alpha\beta$ fraction of the optimal reward, and thus we use the following $(\alpha, \beta)$-approximation regret:

**Definition 1 ($(\alpha, \beta)$-approximation Regret).** *The $T$-round $(\alpha, \beta)$-approximation regret of a learning algorithm $A$ (using an $(\alpha, \beta)$-approximation oracle) for a CMAB-T game instance*

$([m], \mathcal{S}, \mathcal{D}, D^{\mathrm{trig}}, R, D)$ *with* $\boldsymbol{\mu} = \mathbb{E}_{X \sim D}[X]$ *is*

$$Reg^A_{\boldsymbol{\mu}, \alpha, \beta}(T) = T \cdot \alpha \cdot \beta \cdot \mathrm{opt}_{\boldsymbol{\mu}} - \mathbb{E}\left[\sum_{i=1}^{T} R(S^A_t, X^{(t)}, \tau_t)\right] = T \cdot \alpha \cdot \beta \cdot \mathrm{opt}_{\boldsymbol{\mu}} - \mathbb{E}\left[\sum_{i=1}^{T} r_{S^A_t}(\boldsymbol{\mu})\right],$$

where $S^A_t$ is the action $A$ selects in round $t$, and the expectation is taken over the randomness of the environment outcomes $X^{(1)}, \ldots, X^{(T)}$, the triggered sets $\tau_1, \ldots, \tau_T$, as well as the possible randomness of algorithm $A$ itself.

We remark that because probabilistically triggered arms may strongly impact the determination of the best action, but they may be hard to trigger and observe, the regret could be worse and the regret analysis is in general harder than CMAB without probabilistically triggered arms.

The above framework essentially follows [7], but we decouple actions from subsets of arms, allow action space to be infinite, and explicitly model triggered set distribution, which makes the framework more powerful in modeling certain applications (see supplementary material for more discussions).

### 2.1 Examples of CMAB-T: Influence Maximization and Cascading Bandits

In social influence maximization [12], we are given a weighted directed graph $G = (V, E, p)$, where $V$ and $E$ are sets of vertices and edges respectively, and each edge $(u, v)$ is associated with a probability $p(u, v)$. Starting from a seed set $S \subseteq V$, influence propagates in $G$ as follows: nodes in $S$ are activated at time 0, and at time $t \geq 1$, a node $u$ activated in step $t - 1$ has one chance to activate its inactive out-neighbor $v$ with an independent probability $p(u, v)$. The *influence spread* of seed set $S$, $\sigma(S)$, is the expected number of activated nodes after the propagation ends. The offline problem of *influence maximization* is to find at most $k$ seed nodes in $G$ such that the influence spread is maximized. Kempe et al. [12] provide a greedy algorithm with approximation ratio $1 - 1/e - \varepsilon$ and success probability $1 - 1/|V|$, for any $\varepsilon > 0$.

For the online influence maximization bandit [7], the edge probabilities $p(u, v)$'s are unknown and need to be learned over time through repeated influence maximization tasks: in each round $t$, $k$ seed nodes $S_t$ are selected, the influence propagation from $S_t$ is observed, the reward is the number of nodes activated in this round, and one wants to repeat this process to cumulate as much reward as possible. Putting it into the CMAB-T framework, the set of edges $E$ is the set of arms $[m]$, and their outcome distribution $D$ is the joint distribution of $m$ independent Bernoulli distributions with means $p(u, v)$ for all $(u, v) \in E$. Any seed set $S \subseteq V$ with at most $k$ nodes is an action. The triggered arm set $\tau_t$ is the set of edges $(u, v)$ reached by the propagation, that is, $u$ can be reached from $S_t$ by passing through only edges $e \in E$ with $X^{(t)}_e = 1$. In this case, the distribution $D^{\mathrm{trig}}(S_t, X^{(t)})$ degenerates to a deterministic triggered set. The reward $R(S_t, X^{(t)}, \tau_t)$ equals to the number of nodes in $V$ that is reached from $S$ through only edges $e \in E$ with $X^{(t)}_e = 1$, and the expected reward is exactly the influence spread $\sigma(S_t)$. The offline oracle is a $(1 - 1/e - \varepsilon, 1/|V|)$-approximation greedy algorithm. We remark that the general triggered set distribution $D^{\mathrm{trig}}(S_t, X^{(t)})$ (together with infinite action space) can be used to model extended versions of influence maximization, such as randomly selected seed sets in general marketing actions [12] and continuous influence maximization [26] (see supplementary material).

Now let us consider combinatorial cascading bandits [14, 16]. In this case, we have $m$ independent Bernoulli random variables $X_1, \ldots, X_m$ as base arms. An action is to select an ordered sequence from a subset of these arms satisfying certain constraint. Playing this action means that the player reveals the outcomes of the arms one by one following the sequence order until certain stopping condition is satisfied. The feedback is the outcomes of revealed arms and the reward is a function form of these arms. In particular, in the disjunctive form the player stops when the first 1 is revealed and she gains reward of 1, or she reaches the end and gains reward 0. In the conjunctive form, the player stops when the first 0 is revealed (and receives reward 0) or she reaches the end with all 1 outcomes (and receives reward 1). Cascading bandits can be used to model online recommendation and advertising (in the disjunctive form with outcome 1 as a click) or network routing reliability (in the conjunctive form with outcome 0 as the routing edge being broken). It is straightforward to see that cascading bandits fit into the CMAB-T framework: $m$ variables are base arms, ordered sequences are actions, and the triggered set is the prefix set of arms until the stopping condition holds.

---
**Algorithm 1** CUCB with computation oracle.
---
**Input:** $m$, Oracle
  1: For each arm $i$, $T_i \leftarrow 0$ {maintain the total number of times arm $i$ is played so far}
  2: For each arm $i$, $\hat{\mu}_i \leftarrow 1$ {maintain the empirical mean of $X_i$}
  3: **for** $t = 1, 2, 3, \ldots$ **do**
  4:    For each arm $i \in [m]$, $\rho_i \leftarrow \sqrt{\frac{3 \ln t}{2 T_i}}$ {the confidence radius, $\rho_i = +\infty$ if $T_i = 0$}
  5:    For each arm $i \in [m]$, $\bar{\mu}_i = \min\{\hat{\mu}_i + \rho_i, 1\}$ {the upper confidence bound}
  6:    $S \leftarrow \text{Oracle}(\bar{\mu}_1, \ldots, \bar{\mu}_m)$
  7:    Play action $S$, which triggers a set $\tau \subseteq [m]$ of base arms with feedback $X_i^{(t)}$'s, $i \in \tau$
  8:    For every $i \in \tau$, update $T_i$ and $\hat{\mu}_i$: $T_i = T_i + 1$, $\hat{\mu}_i = \hat{\mu}_i + (X_i^{(t)} - \hat{\mu}_i)/T_i$
  9: **end for**
---

## 3 Triggering Probability Modulated Condition

Chen et al. [7] use two conditions to guarantee the theoretical regret bounds. The first one is monotonicity, which we also use in this paper, and is restated below.

**Condition 1 (Monotonicity).** *We say that a CMAB-T problem instance satisfies* monotonicity, *if for any action $S \in \mathcal{S}$, for any two distributions $D, D' \in \mathcal{D}$ with expectation vectors $\boldsymbol{\mu} = (\mu_1, \ldots, \mu_m)$ and $\boldsymbol{\mu}' = (\mu_1', \ldots, \mu_m')$, we have $r_S(\boldsymbol{\mu}) \leq r_S(\boldsymbol{\mu}')$ if $\mu_i \leq \mu_i'$ for all $i \in [m]$.*

The second condition is bounded smoothness. One key contribution of our paper is to properly strengthen the original bounded smoothness condition in [7] so that we can both get rid of the undesired $1/p^*$ term in the regret bound and guarantee that many CMAB problems still satisfy the conditions. Our important change is to use triggering probabilities to modulate the condition, and thus we call such conditions *triggering probability modulated (TPM)* conditions. The key point of TPM conditions is including the triggering probability in the condition. We use $p_i^{D,S}$ to denote the probability that action $S$ triggers arm $i$ when the environment instance is $D$. With this definition, we can also technically define $p^*$ as $p^* = \inf_{i \in [m], S \in \mathcal{S}, p_i^{D,S} > 0} p_i^{D,S}$. In this section, we further use 1-norm based conditions instead of the infinity-norm based condition in [7], since they lead to better regret bounds for the influence maximization and cascading bandits.

**Condition 2 (1-Norm TPM Bounded Smoothness).** *We say that a CMAB-T problem instance satisfies 1-norm TPM bounded smoothness, if there exists $B \in \mathbb{R}^+$ (referred as the* bounded smoothness constant*) such that, for any two distributions $D, D' \in \mathcal{D}$ with expectation vectors $\boldsymbol{\mu}$ and $\boldsymbol{\mu}'$, and any action $S$, we have $|r_S(\boldsymbol{\mu}) - r_S(\boldsymbol{\mu}')| \leq B \sum_{i \in [m]} p_i^{D,S} |\mu_i - \mu_i'|$.*

Note that the corresponding non-TPM version of the above condition would remove $p_i^{D,S}$ in the above condition, which is a generalization of the linear condition used in linear bandits [15]. Thus, the TPM version is clearly stronger than the non-TPM version (when the bounded smoothness constants are the same). The intuition of incorporating the triggering probability $p_i^{D,S}$ to modulate the 1-norm condition is that, when an arm $i$ is unlikely triggered by action $S$ (small $p_i^{D,S}$), the importance of arm $i$ also diminishes in that a large change in $\mu_i$ only causes a small change in the expected reward $r_S(\boldsymbol{\mu})$. This property sounds natural in many applications, and it is important for bandit learning — although an arm $i$ may be difficult to observe when playing $S$, it is also not important to the expected reward of $S$ and thus does not need to be learned as accurately as others more easily triggered by $S$.

## 4 CUCB Algorithm and Regret Bound with TPM Bounded Smoothness

We use the same CUCB algorithm as in [7] (Algorithm 1). The algorithm maintains the empirical estimate $\hat{\mu}_i$ for the true mean $\mu_i$, and feed the upper confidence bound $\bar{\mu}_i$ to the offline oracle to obtain the next action $S$ to play. The upper confidence bound $\bar{\mu}_i$ is large if arm $i$ is not triggered often ($T_i$ is small), providing optimistic estimates for less observed arms. We next provide its regret bound.

**Definition 2 (Gap).** *Fix a distribution $D$ and its expectation vector $\boldsymbol{\mu}$. For each action $S$, we define the gap $\Delta_S = \max(0, \alpha \cdot \mathrm{opt}_{\boldsymbol{\mu}} - r_S(\boldsymbol{\mu}))$. For each arm $i$, we define*

$$\Delta^i_{\min} = \inf_{S \in \mathcal{S}: p_i^{D,S} > 0, \Delta_S > 0} \Delta_S, \qquad \Delta^i_{\max} = \sup_{S \in \mathcal{S}: p_i^{D,S} > 0, \Delta_S > 0} \Delta_S.$$

*As a convention, if there is no action $S$ such that $p_i^{D,S} > 0$ and $\Delta_S > 0$, we define $\Delta^i_{\min} = +\infty$, $\Delta^i_{\max} = 0$. We define $\Delta_{\min} = \min_{i \in [m]} \Delta^i_{\min}$, and $\Delta_{\max} = \max_{i \in [m]} \Delta^i_{\max}$.*

Let $\tilde{S} = \{i \in [m] \mid p_i^{\boldsymbol{\mu},S} > 0\}$ be the set of arms that could be triggered by $S$. Let $K = \max_{S \in \mathcal{S}} |\tilde{S}|$. For convenience, we use $\lceil x \rceil_0$ to denote $\max\{\lceil x \rceil, 0\}$ for any real number $x$.

**Theorem 1.** *For the CUCB algorithm on a CMAB-T problem instance that satisfies monotonicity (Condition 1) and 1-norm TPM bounded smoothness (Condition 2) with bounded smoothness constant $B$, (1) if $\Delta_{\min} > 0$, we have distribution-dependent bound*

$$Reg_{\boldsymbol{\mu},\alpha,\beta}(T) \leq \sum_{i \in [m]} \frac{576 B^2 K \ln T}{\Delta^i_{\min}} + \sum_{i \in [m]} \left( \left\lceil \log_2 \frac{2BK}{\Delta^i_{\min}} \right\rceil_0 + 2 \right) \cdot \frac{\pi^2}{6} \cdot \Delta_{\max} + 4Bm; \quad (1)$$

*(2) we have distribution-independent bound*

$$Reg_{\boldsymbol{\mu},\alpha,\beta}(T) \leq 12B\sqrt{mKT \ln T} + \left( \left\lceil \log_2 \frac{T}{18 \ln T} \right\rceil_0 + 2 \right) \cdot m \cdot \frac{\pi^2}{6} \cdot \Delta_{\max} + 2Bm. \quad (2)$$

For the above theorem, we remark that the regret bounds are tight (up to a $O(\sqrt{\log T})$ factor in the case of distribution-independent bound) base on a lower bound result in [15]. More specifically, Kveton et al. [15] show that for linear bandits (a special class of CMAB-T without probabilistic triggering), the distribution-dependent regret is lower bounded by $\Omega(\frac{(m-K)K}{\Delta} \log T)$, and the distribution-independent regret is lower bounded by $\Omega(\sqrt{mKT})$ when $T \geq m/K$, for some instance where $\Delta^i_{\min} = \Delta$ for all $i \in [m]$ and $\Delta^i_{\min} < \infty$. Comparing with our regret upper bound in the above theorem, (a) for distribution-dependent bound, we have the regret upper bound $O(\frac{(m-K)K}{\Delta} \log T)$ since for that instance $B = 1$ and there are $K$ arms with $\Delta^i_{\min} = \infty$, so tight with the lower bound in [15]; and (b) for distribution-independent bound, we have the regret upper bound $O(\sqrt{mKT \log T})$, tight to the lower bound up to a $O(\sqrt{\log T})$ factor, same as the upper bound for the linear bandits in [15]. This indicates that parameters $m$ and $K$ appeared in the above regret bounds are all needed. As for parameter $B$, we can view it simply as a scaling parameter. If we scale the reward of an instance to $B$ times larger than before, certainly, the regret is $B$ times larger. Looking at the distribution-dependent regret bound (Eq. (1)), $\Delta^i_{\min}$ would also be scaled by a factor of $B$, canceling one $B$ factor from $B^2$, and $\Delta_{\max}$ is also scaled by a factor of $B$, and thus the regret bound in Eq. (1) is also scaled by a factor of $B$. In the distribution-independent regret bound (Eq. (2)), the scaling of $B$ is more direct. Therefore, we can see that all parameters $m$, $K$, and $B$ appearing in the above regret bounds are needed. Finally, we remark that the TPM Condition 2 can be refined such that $B$ is replaced by arm-dependent $B_i$ that is moved inside the summation, and $B$ in Theorem 1 is replaced with $B_i$ accordingly. See the supplementary material for details.

## 4.1 Novel Ideas in the Regret Analysis

Due to the space limit, the full proof of Theorem 1 is moved to the supplementary material. Here we briefly explain the novel aspects of our analysis that allow us to achieve new regret bounds and differentiate us from previous analyses such as the ones in [7] and [16, 15].

We first give an intuitive explanation on how to incorporate the TPM bounded smoothness condition to remove the factor $1/p^*$ in the regret bound. Consider a simple illustrative example of two actions $S_0$ and $S$, where $S_0$ has a fixed reward $r_0$ as a reference action, and $S$ has a stochastic reward depending on the outcomes of its triggered base arms. Let $\tilde{S}$ be the set of arms that can be triggered by $S$. For $i \in \tilde{S}$, suppose $i$ can be triggered by action $S$ with probability $p_i^S$, and its true mean is $\mu_i$ and its empirical mean at the end of round $t$ is $\hat{\mu}_{i,t}$. The analysis in [7] would need a property that, if for all $i \in \tilde{S}$ $|\hat{\mu}_{i,t} - \mu_i| \leq \delta_i$ for some properly defined $\delta_i$, then $S$ no longer generates regrets. The analysis would conclude that arm $i$ needs to be triggered $\Theta(\log T/\delta_i^2)$ times for the above condition

to happen. Since arm $i$ is only triggered with probability $p_i^S$, it means action $S$ may need to be played $\Theta(\log T/(p_i^S \delta_i^2))$ times. This is the essential reason why the factor $1/p^*$ appears in the regret bound.

Now with the TPM bounded smoothness, we know that the impact of $|\hat{\mu}_{i,t} - \mu_i| \leq \delta_i$ to the difference in the expected reward is only $p_i^S \delta_i$, or equivalently, we could relax the requirement to $|\hat{\mu}_{i,t} - \mu_i| \leq \delta_i/p_i^S$ to achieve the same effect as in the previous analysis. This translates to the result that action $S$ would generate regret in at most $O(\log T/(p_i^S(\delta_i/p_i^S)^2)) = O(p_i^S \log T/\delta_i^2)$ rounds.

We then need to handle the case when we have multiple actions that could trigger arm $i$. The simple addition of $\sum_{S:p_i^S > 0} p_i^S \log T/\delta_i^2$ is not feasible since we may have exponentially or even infinitely many such actions. Instead, we introduce the key idea of *triggering probability groups*, such that the above actions are divided into groups by putting their triggering probabilities $p_i^S$ into geometrically separated bins: $(1/2, 1], (1/4, 1/2] \ldots, (2^{-j}, 2^{-j+1}], \ldots$. The actions in the same group would generate regret in at most $O(2^{-j+1} \log T/\delta_i^2)$ rounds with a similar argument, and summing up together, they could generate regret in at most $O(\sum_j 2^{-j+1} \log T/\delta_i^2) = O(\log T/\delta_i^2)$ rounds. Therefore, the factor of $1/p_i^S$ or $1/p^*$ is completely removed from the regret bound.

Next, we briefly explain our idea to achieve the improved bound over the linear bandit result in [15]. The key step is to bound regret $\Delta_{S_t}$ generated in round $t$. By a derivation similar to [15, 7] together with the 1-norm TPM bounded smoothness condition, we would obtain that $\Delta_{S_t} \leq B \sum_{i \in \tilde{S}_t} p_i^{D,S_t}(\bar{\mu}_{i,t} - \mu_i)$ with high probability. The analysis in [15] would analyze the errors $|\bar{\mu}_{i,t} - \mu_i|$ by a cascade of infinitely many sub-cases of whether there are $x_j$ arms with errors larger than $y_j$ with decreasing $y_j$, but it may still be loose. Instead we directly work on the above summation. Naive bounding the about error summation would not give a $O(\log T)$ bound because there could be too many arms with small errors. Our trick is to use a *reverse amortization*: we cumulate small errors on many sufficiently sampled arms and treat them as errors of insufficiently sample arms, such that an arm sampled $O(\log T)$ times would not contribute toward the regret. This trick tightens our analysis and leads to significantly improved constant factors.

## 4.2 Applications to Influence Maximization and Combinatorial Cascading Bandits

The following two lemmas show that both the cascading bandits and the influence maximization bandit satisfy the TPM condition.

**Lemma 1.** *For both disjunctive and conjunctive cascading bandit problem instances, 1-norm TPM bounded smoothness (Condition 2) holds with bounded smoothness constant $B = 1$.*

**Lemma 2.** *For the influence maximization bandit problem instances, 1-norm TPM bounded smoothness (Condition 2) holds with bounded smoothness constant $B = \tilde{C}$, where $\tilde{C}$ is the largest number of nodes any node can reach in the directed graph $G = (V, E)$.*

The proof of Lemma 1 involves a technique called *bottom-up modification*. Each action in cascading bandits can be viewed as a chain from top to bottom. When changing the means of arms below, the triggering probability of arms above is not changed. Thus, if we change $\boldsymbol{\mu}$ to $\boldsymbol{\mu}'$ backwards, the triggering probability of each arm is unaffected before its expectation is changed, and when changing the mean of an arm $i$, the expected reward of the action is at most changed by $p_i^{D,S}|\mu_i' - \mu_i|$.

The proof of Lemma 2 is more complex, since the bottom-up modification does not work directly on graphs with cycles. To circumvent this problem, we develop an *influence tree decomposition* technique as follows. First, we order all influence paths from the seed set $S$ to a target $v$. Second, each edge is independently sampled based on its edge probability to form a random *live-edge graph*. Third, we divide the reward portion of activating $v$ among all paths from $S$ to $v$: for each live-edge graph $L$ in which $v$ is reachable from $S$, assign the probability of $L$ to the first path from $S$ to $v$ in $L$ according to the path total order. Finally, we compose all the paths from $S$ to $v$ into a tree with $S$ as the root and copies of $v$ as the leaves, so that we can do bottom-up modification on this tree and properly trace the reward changes based on the reward division we made among the paths.

## 4.3 Discussions and Comparisons

We now discuss the implications of Theorem 1 together with Lemmas 1 and 2 by comparing them with several existing results.

**Comparison with [7] and CMAB with $\infty$-norm bounded smoothness conditions.** Our work is a direct adaption of the study in [7]. Comparing with [7], we see that the regret bounds in Theorem 1 are not dependent on the inverse of triggering probabilities, which is the main issue in [7]. When applied to influence maximization bandit, our result is strictly stronger than that of [7] in two aspects: (a) we remove the factor of $1/p^*$ by using the TPM condition; (b) we reduce a factor of $|E|$ and $\sqrt{|E|}$ in the dominant terms of distribution-dependent and -independent bounds, respectively, due to our use of 1-norm instead of $\infty$-norm conditions used in Chen et al. [7]. In the supplementary material, we further provide the corresponding $\infty$-norm TPM bounded smoothness conditions and the regret bound results, since in general the two sets of results do not imply each other.

**Comparison with [25] on influence maximization bandits.** Conceptually, our work deals with the general CMAB-T framework with influence maximization and combinatorial cascading bandits as applications, while Wen et al. [25] only work on influence maximization bandit. Wen et al. [25] further study a generalization of linear transformation of edge probabilities, which is orthogonal to our current study, and could be potentially incorporated into the general CMAB-T framework. Technically, both studies eliminate the exponential factor $1/p^*$ in the regret bound. Comparing the rest terms in the regret bounds, our regret bound depends on a topology dependent term $\tilde{C}$ (Lemma 2), while their bound depends on a complicated term $C_*$, which is related to both topology and edge probabilities. Although in general it is hard to compare the regret bounds, for the several graph families for which Wen et al. [25] provide concrete topology-dependent regret bounds, our bounds are always better by a factor from $O(\sqrt{k})$ to $O(|V|)$, where $k$ is the number of seeds selected in each round and $V$ is the node set in the graph. This indicates that, in terms of characterizing the topology effect on the regret bound, our simple complexity term $\tilde{C}$ is more effective than their complicated term $C_*$. See the supplementary material for the detailed table of comparison.

**Comparison with [16] on combinatorial cascading bandits** By Lemma 1, we can apply Theorem 1 to combinatorial conjunctive and disjunctive cascading bandits with bounded smoothness constant $B = 1$, achieving $O(\sum \frac{1}{\Delta^i_{\min}} K \log T)$ distribution-dependent, and $O(\sqrt{mKT \log T})$ distribution-independent regret. In contrast, besides having exactly these terms, the results in [16] have an extra factor of $1/f^*$, where $f^* = \prod_{i \in S^*} p(i)$ for conjunctive cascades, and $f^* = \prod_{i \in S^*} (1 - p(i))$ for disjunctive cascades, with $S^*$ being the optimal solution and $p(i)$ being the probability of success for item (arm) $i$. For conjunctive cascades, $f^*$ could be reasonably close to 1 in practice as argued in [16], but for disjunctive cascades, $f^*$ could be exponentially small since items in optimal solutions typically have large $p(i)$ values. Therefore, our result completely removes the dependency on $1/f^*$ and is better than their result. Moreover, we also have much smaller constant factors owing to the new reverse amortization method described in Section 4.1.

**Comparison with [15] on linear bandits.** When there is no probabilistically triggered arms (i.e. $p^* = 1$), Theorem 1 would have tighter bounds since some analysis dealing with probabilistic triggering is not needed. In particular, in Eq. (1) the leading constant $624$ would be reduced to $48$, the $\lceil \log_2 x \rceil_0$ term is gone, and $6Bm$ becomes $2Bm$; in Eq. (2) the leading constant $50$ is reduced to $14$, and the other changes are the same as above (see the supplementary material). The result itself is also a new contribution, since it generalizes the linear bandit of [15] to general 1-norm conditions with matching regret bounds, while significantly reducing the leading constants (their constants are $534$ and $47$ for distribution-dependent and independent bounds, respectively). This improvement comes from the new reversed amortization method described in Section 4.1.

## 5 Lower Bound of the General CMAB-T Model

In this section, we show that there exists some CMAB-T problem instance such that the regret bound in [7] is tight, i.e. the factor $1/p^*$ in the distribution-dependent bound and $\sqrt{1/p^*}$ in the distribution-independent bound are unavoidable, where $p^*$ is the minimum positive probability that any base arm $i$ is triggered by any action $S$. It also implies that the TPM bounded smoothness may not be applied to all CMAB-T instances.

For our purpose, we only need a simplified version of the bounded smoothness condition of [7] as below: There exists a bounded smoothness constant $B$ such that, for every action $S$ and every pair of mean outcome vectors $\boldsymbol{\mu}$ and $\boldsymbol{\mu}'$, we have $|r_S(\boldsymbol{\mu}) - r_S(\boldsymbol{\mu}')| \le B \max_{i \in \tilde{S}} |\mu_i - \mu_i'|$, where $\tilde{S}$ is the set of arms that could possibly be triggered by $S$.

We prove the lower bounds using the following CMAB-T problem instance $([m], \mathcal{S}, \mathcal{D}, D^{\mathrm{trig}}, R)$. For each base arm $i \in [m]$, we define an action $S_i$, with the set of actions $\mathcal{S} = \{S_1, \ldots, S_m\}$. The family of distributions $\mathcal{D}$ consists of distributions generated by every $\boldsymbol{\mu} \in [0, 1]^m$ such that the arms are independent Bernoulli variables. When playing action $S_i$ in round $t$, with a fixed probability $p$, arm $i$ is triggered and its outcome $X_i^{(t)}$ is observed, and the reward of playing $S_i$ is $p^{-1} X_i^{(t)}$; otherwise with probability $1 - p$ no arm is triggered, no feedback is observed and the reward is $0$. Following the CMAB-T framework, this means that $D^{\mathrm{trig}}(S_i, X)$, as a distribution on the subsets of $[m]$, is either $\{i\}$ with probability $p$ or $\varnothing$ with probability $1 - p$, and the reward $R(S_i, X, \tau) = p^{-1} X_i \cdot \mathbb{I}\{\tau = \{i\}\}$. The expected reward $r_{S_i}(\boldsymbol{\mu}) = \mu_i$. So this instance satisfies the above bounded smoothness with constant $B = 1$. We denote the above instance as FTP($p$), standing for fixed triggering probability instance. This instance is similar with position-based model [17] with only one position, while the feedback is different. For the FTP($p$) instance, we have $p^* = p$ and $r_{S_i}(\boldsymbol{\mu}) = p \cdot p^{-1} \mu_i = \mu_i$. Then applying the result in [7], we have distributed-dependent upper bound $O(\sum_i \frac{1}{p \Delta_{\min}^i} \log T)$ and distribution-independent upper bound $O(\sqrt{p^{-1} mT \log T})$.

We first provide the distribution-independent lower bound result.

**Theorem 2.** *Let $p$ be a real number with $0 < p < 1$. Then for any CMAB-T algorithm $A$, if $T \geq 6p^{-1}$, there exists a CMAB-T environment instance $D$ with mean $\boldsymbol{\mu}$ such that on instance FTP($p$),*

$$Reg_{\boldsymbol{\mu}}^A(T) \geq \frac{1}{170} \sqrt{\frac{mT}{p}}.$$

The proof of the above and the next theorem are all based on the results for the classical MAB problems. Comparing to the upper bound $O(\sqrt{p^{-1} mT \log T})$. obtained from [7], Theorem 2 implies that the regret upper bound of CUCB in [7] is tight up to a $O(\sqrt{\log T})$ factor. This means that the $1/p^*$ factor in the regret bound of [7] cannot be avoided in the general class of CMAB-T problems.

Next we give the distribution-dependent lower bound. For a learning algorithm, we say that it is *consistent* if, for every $\boldsymbol{\mu}$, every non-optimal arm is played $o(T^a)$ times in expectation, for any real number $a > 0$. Then we have the following distribution-dependent lower bound.

**Theorem 3.** *For any consistent algorithm $A$ running on instance FTP($p$) and $\mu_i < 1$ for every arm $i$, we have*

$$\liminf_{T \to +\infty} \frac{Reg_{\boldsymbol{\mu}}^A(T)}{\ln T} \geq \sum_{i: \mu_i < \mu^*} \frac{p^{-1} \Delta_i}{\mathrm{kl}(\mu_i, \mu^*)},$$

*where $\mu^* = \max_i \mu_i$, $\Delta_i = \mu^* - \mu_i$, and $\mathrm{kl}(\cdot, \cdot)$ is the Kullback-Leibler divergence function.*

Again we see that the distribution-dependent upper bound obtained from [7] asymptotically match the lower bound above. Finally, we remark that even if we rescale the reward from $[1, 1/p]$ back to $[0, 1]$, the corresponding scaling factor $B$ would become $p$, and thus we would still obtain the conclusion that the regret bounds in [7] is tight (up to a $O(\sqrt{\log T})$ factor), and thus $1/p^*$ is in general needed in those bounds.

## 6  Conclusion and Future Work

In this paper, we propose the TPM bounded smoothness condition, which conveys the intuition that an arm difficult to trigger is also less important in determining the optimal solution. We show that this condition is essential to guarantee low regret, and prove that important applications, such as influence maximization bandits and combinatorial cascading bandits all satisfy this condition.

There are several directions one may further pursue. One is to improve the regret bound for some specific problems. For example, for the influence maximization bandit, can we give a better algorithm or analysis to achieve a better regret bound than the one provided by the general TPM condition? Another direction is to look into other applications with probabilistically triggered arms that may not satisfy the TPM condition or need other conditions to guarantee low regret. Combining the current CMAB-T framework with the linear generalization as in [25] to achieve scalable learning result is also an interesting direction.

## Acknowledgment

Wei Chen is partially supported by the National Natural Science Foundation of China (Grant No. 61433014).

## Footnotes

[1] The factor of $1/f^*$ used for the combinatorial disjunctive cascading bandits in [16] is essentially $1/p^*$.

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
