[Supplementary Material]

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

[2]The result in the book by [19] (Theorem 4.5 together with Exercise 4.7) only covers the case where random variables $X_i$'s are independent. However the result can be easily generalized to our case with an almost identical proof. The only main change is to replace $\mathbb{E}\left[e^{t(\sum_{j=1}^{i-1} X_j + X_i)}\right] = \mathbb{E}\left[e^{t\sum_{j=1}^{i-1} X_j}\right] \mathbb{E}\left[e^{tX_i}\right]$ with $\mathbb{E}\left[e^{t(\sum_{j=1}^{i-1} X_j + X_i)}\right] = \mathbb{E}\left[e^{t\sum_{j=1}^{i-1} X_j} \mathbb{E}\left[e^{tX_i} \mid X_1, \ldots, X_{i-1}\right]\right]$.

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

# Supplementary Materials

## A  Model Discussions

### A.1  Comparison with the framework of [7]

The CMAB-T framework described above essentially follows the framework of [7], but with the following noticeable differences. First, we refer to $S$ as an abstract action from an action space $\mathcal{S}$, while in [7], $S$ is referred to as a super arm, which is a subset of base arms $[m]$. In the case of CMAB without probabilistically triggered arms, we can simply let every super arm $S$ be an action, and $\tau(S, X) = S$, meaning that playing super arm $S$ deterministically triggers all and only base arms in $S \subseteq [m]$. Second, we explicitly allows action space to be infinite or even continuous space, while in [7], the action space is the subsets of base arms and thus is finite. We will see later that the infinite action space does not make essential difference in the analysis. Third, for probabilistically triggered arms, we explicitly use $\tau(S, X)$ to model them, and allows $\tau(S, X)$ to have additional randomness besides the randomness of $X$. In [7], probabilistic triggering is explained as further base arms being triggered based on the outcomes of previously triggered base arms, and to model certain triggering structure or additional randomness in triggering an arm, dummy base arms need to be added. However, this may require introducing a large number of dummy base arms. For example, for the cascading bandits, to specify the order of the cascade sequence, we need to add dummy base arms corresponding to every possible order of the base arms. Moreover, $\tau(S, X)$ cleanly separates the randomness known to the player from the unknown randomness from the environment outcome. For example, in the discount-based continuous influence maximization [26], $\tau(c, X)$ includes the randomness of activating the seed set from the discount vector $c$ given by $\eta_i$'s, which are known to the player. In contrast, the distribution of $X_{(u,v)}$, namely probability $p(u, v)$ on edges are unknown and need to be learned. In this case, if we use dummy base arms to model such additional triggering behavior from marketing actions to seed sets, these dummy base arms will be mixed together with edge base arms for which the learning algorithm need to learn, unless further distinction is made.

Therefore, we believe that our current adaptation CMAB-T provides a cleaner framework and is more easily to be applied to various problem instances. We remark that all the analysis and results in [7] remain unchanged with our current adaptation.

### A.2  Modeling general marketing actions in influence maximization

Note that we can also use randomized $\tau(S, X)$ to model some extended versions of influence maximization. For example, general marketing actions are proposed in [12] and continuous discount actions are proposed in [26], both allowing activating seed nodes with a probability depending on the marketing intensity on the node. In particular, an action in the discount-based continuous influence maximization in [26] is a vector $c = (c_1, c_2, \ldots, c_n)$, where $c_i \in [0, 1]$ is the discount to be given to node $i$. Discount $c_i$ is translated to probability $\eta_i(c_i)$ that node $i$ is activated as a seed, where $\eta_i(\cdot)$ is a monotonically non-decreasing function with $\eta_i(0) = 0$ and $\eta_i(1) = 1$. In this case, the probabilistic triggering function $\tau(c, X)$ includes the randomness from $c$ to seed activations based on $\eta_i$'s, beyond the randomness of $X$. That is, even when $c$ and $X$ are fixed, $\tau(c, X)$ is still a random set. We further remark that in this case, the action space of all discount vectors is a continuous and infinite space, which is allowed in our adapted CMAB-T model.

## B  Main Regret Analysis (Proofs Related to Theorem 1)

### B.1  Basics of CMAB-T problems

We utilize the following well known tail bound in our analysis.

**Fact 1 (Hoeffding's Inequality [11]).** *Let $X_1, \cdots, X_n$ be independent and identically distributed random variables with common support $[0, 1]$ and mean $\mu$. Let $Y = X_1 + \cdots + X_n$. Then for all $\delta \geq 0$,*

$$\Pr\{|Y - n\mu| \geq \delta\} \leq 2e^{-2\delta^2/n}.$$

**Fact 2 (Multiplicative Chernoff Bound [19]).** [2] *Let $X_1, \cdots, X_n$ be Bernoulli random variables taking values from $\{0,1\}$, and $\mathbb{E}[X_t | X_1, \cdots, X_{t-1}] \geq \mu$ for every $t \leq n$. Let $Y = X_1 + \cdots + X_n$. Then for all $0 < \delta < 1$,*

$$\Pr\{Y \leq (1-\delta)n\mu\} \leq e^{-\frac{\delta^2 n\mu}{2}}.$$

We introduce the following definition to assist our analysis.

**Definition 3 (Event-Filtered Regret).** *For any series of events $\{\mathcal{E}_t\}_{t \geq 1}$ indexed by round number $t$, we define $Reg^A_{\boldsymbol{\mu}, \alpha}(T, \{\mathcal{E}_t\}_{t\geq 1})$ as the regret filtered by events $\{\mathcal{E}_t\}_{t \geq 1}$, that is, regret is only counted in round $t$ if $\mathcal{E}_t$ happens in round $t$. Formally,*

$$Reg^A_{\boldsymbol{\mu}, \alpha}(T, \{\mathcal{E}_t\}_{t\geq 1}) = \mathbb{E}\left[\sum_{t=1}^{T} \mathbb{I}(\mathcal{E}_t)(\alpha \cdot \mathrm{opt}_{\boldsymbol{\mu}} - r_{\boldsymbol{\mu}}(S_t^A))\right].$$

*For convenience, $A$, $\alpha$, $\boldsymbol{\mu}$ and/or $T$ can be omitted when the context is clear, and we simply use $Reg^A_{\boldsymbol{\mu}, \alpha}(T, \mathcal{E}_t)$ instead of $Reg^A_{\boldsymbol{\mu}, \alpha}(T, \{\mathcal{E}_t\}_{t\geq 1})$.*

The following definition describes an unlikely event that $\hat{\mu}_{i,t-1}$ is not as accurate as expected.

**Definition 4.** *We say that the* sampling is nice *at the beginning of round $t$ if for every arm $i \in [m]$, $|\hat{\mu}_{i,t-1} - \mu_i| < \rho_{i,t}$, where $\rho_{i,t} = \sqrt{\frac{3\ln t}{2T_{i,t-1}}}$ in round $t$. Let $\mathcal{N}_t^s$ be such event.*

**Lemma 3.** *For each round $t \geq 1$, $\Pr\{\neg\mathcal{N}_t^s\} \leq 2mt^{-2}$.*

*Proof.* For each round $t \geq 1$, we have

$$\Pr\{\neg\mathcal{N}_t^s\} = \Pr\left\{\exists i \in [m], |\hat{\mu}_{i,t-1} - \mu_i| \geq \sqrt{\frac{3\ln t}{2T_{i,t-1}}}\right\}$$

$$\leq \sum_{i\in[m]} \Pr\left\{|\hat{\mu}_{i,t-1} - \mu_i| \geq \sqrt{\frac{3\ln t}{2T_{i,t-1}}}\right\}.$$

$$= \sum_{i\in[m]} \sum_{k=1}^{t-1} \Pr\left\{T_{i,t-1} = k, |\hat{\mu}_{i,t-1} - \mu_i| \geq \sqrt{\frac{3\ln t}{2T_{i,t-1}}}\right\}. \tag{3}$$

When $T_{i,t-1} = k$, $\hat{\mu}_{i,t-1}$ is the average of $k$ i.i.d. random variables $X_i^{[1]}, \ldots, X_i^{[k]}$, where $X_i^{[j]}$ is the outcome of arm $i$ when it is triggered for the $j$-th time during the execution. That is, $\hat{\mu}_{i,t-1} = \sum_{j=1}^{k} X_i^{[j]}/k$. Then we have

$$\Pr\left\{T_{i,t-1} = k, |\hat{\mu}_{i,t-1} - \mu_i| \geq \sqrt{\frac{3\ln t}{2T_{i,t-1}}}\right\} = \Pr\left\{T_{i,t-1} = k, \left|\sum_{j=1}^{k} X_i^{[j]}/k - \mu_i\right| \geq \sqrt{\frac{3\ln t}{2k}}\right\}$$

$$\leq \Pr\left\{\left|\sum_{j=1}^{k} X_i^{[j]} - k\mu_i\right| \geq \sqrt{\frac{3k\ln t}{2}}\right\} \leq 2t^{-3}, \tag{4}$$

where the last inequality uses the Hoeffding's Inequality (Fact 1). Combining Inequalities (3) and (4), we thus prove the lemma. $\square$

**Definition 5 (Triggering probability (TP) group).** *Let $i$ be an arm and $j$ be a positive natural number, define the triggering probability group (of actions)*

$$\mathcal{S}_{i,j}^D = \{S \in \mathcal{S} \mid 2^{-j} < p_i^{D,S} \leq 2^{-j+1}\}.$$

*Notice $\{\mathcal{S}_{i,j}^D\}_{j \geq 1}$ forms a partition of $\{S \in \mathcal{S} \mid p_i^{D,S} > 0\}$.*

**Definition 6 (Counter).** *For each TP group $\mathcal{S}_{i,j}$, we define a corresponding counter $N_{i,j}$. In a run of a learning algorithm, the counters are maintained in the following manner. All the counters are initialized to $0$. In each round $t$, if the action $S_t$ is chosen, then update $N_{i,j}$ to $N_{i,j}+1$ for every $(i,j)$ that $S_t \in \mathcal{S}_{i,j}^D$. Denote $N_{i,j}$ at the end of round $t$ with $N_{i,j,t}$. In other words, we can define the counters with the recursive equation below:*

$$N_{i,j,t} = \begin{cases} 0, & \text{if } t = 0 \\ N_{i,j,t-1}+1, & \text{if } t > 0, S_t \in \mathcal{S}_{i,j}^D \\ N_{i,j,t-1}, & \text{otherwise.} \end{cases}$$

**Definition 7.** *Given a series of integers $\{j_{\max}^i\}_{i\in[m]}$, we say that the* triggering is nice *at the beginning of round $t$ (with respect to $j_{\max}^i$), if for every TP group (Definition 5) identified by arm $i$ and $1 \le j \le j_{\max}^i$, as long as $\sqrt{\frac{6\ln t}{\frac{1}{3}N_{i,j,t-1}\cdot 2^{-j}}} \le 1$, there is $T_{i,t-1} \ge \frac{1}{3}N_{i,j,t-1}\cdot 2^{-j}$. We denote this event with $\mathcal{N}_t^t$. It implies*

$$\rho_{i,t} = \sqrt{\frac{3\ln t}{2T_{i,t-1}}} \le \sqrt{\frac{3\ln t}{2\cdot\frac{1}{3}N_{i,j,t-1}\cdot 2^{-j}}}.$$

**Lemma 4.** *For a series of integers $\{j_{\max}^i\}_{i\in[m]}$, $\Pr\{\neg\mathcal{N}_t^t\} \le \sum_{i\in[m]} j_{\max}^i t^{-2}$ for every round $t \ge 1$.*

*Proof.* We prove this lemma by showing $\Pr\{N_{i,j,t-1} = s, T_{i,t-1} \le \frac{1}{3}N_{i,j,t-1}\cdot 2^{-j}\} \le t^{-3}$, for $0 \le s \le t-1$ and $\sqrt{\frac{6\ln t}{s\cdot 2^{-j}}} \le 1$. Let $t_k$ be the round that $N_{i,j}$ is increased for the $k$-th time, for $1 \le k \le s$. Let $Y_k = \mathbb{I}\{i \in \tau_{t_k}\}$ be a Bernoulli variable, that is, $i$ is triggered in round $t_k$. When fixing the action $S_{t_k}$, $Y_k$ is independent from $Y_1, \ldots, Y_{k-1}$. Since $S_{t_k} \in \mathcal{S}_{i,j}$, $\mathbb{E}[Y_k \mid Y_1, \ldots, Y_{k-1}] \ge 2^{-j}$. Let $Z = Y_1 + \cdots + Y_s$. By multiplicative Chernoff bound (Fact 2), we have

$$\Pr\left\{Z < \frac{1}{3}s\cdot 2^{-j}\right\} < \exp\left(-\left(\frac{2}{3}\right)^2 18\ln t/2\right) < \exp(-3\ln t) = t^{-3}.$$

By definition of $T_i$, there is $T_{i,t-1} \ge Z$. So $\Pr\{N_{i,j,t-1} = s, T_{i,t-1} \le \frac{1}{3}N_{i,j,t-1}\cdot 2^{-j}\} \le t^{-3}$. By taking $i$ over $[m]$, $j$ over $1, \ldots, j_{\max}^i$, $s$ over $0, \ldots, t-1$, the lemma holds. $\square$

## B.2 The Case of No Probabilistically Triggered Arms

In this section, we state and prove a theorem for the case of no probabilistically triggered arms, i.e. $p^* = 1$, when the CMAB-T instance satisfies the 1-norm (non-TPM) bounded smoothness condition below.

**Condition 3 (1-Norm Bounded Smoothness).** *We say that a CMAB-T problem instance satisfies 1-norm bounded smoothness, if there exists a bounded smoothness constant $B \in \mathbb{R}^+$ such that, for any two distributions $D, D' \in \mathcal{D}$ with expectation vectors $\boldsymbol{\mu}$ and $\boldsymbol{\mu}'$, and any action $S$, we have $|r_S(\boldsymbol{\mu}) - r_S(\boldsymbol{\mu}')| \le B \sum_{i\in\tilde{S}} |\mu_i - \mu_i'|$, where $\tilde{S}$ is the set of arms that are triggered by $S$.*

As discussed in the main text, this theorem provides better bounds than Theorem 1 with probabilistically triggered arms. Its proof is also simpler, so the readers could choose to either get oneself familiar with the analysis with this proof first, or directly jump to the next section for the proof of Theorem 1.

**Theorem 4.** *For the CUCB algorithm on a CMAB (without triggering, i.e. $p^* = 1$) problem that satisfies 1-norm bounded smoothness (Condition 3) with bounded smoothness constant $B$,*

1. *if $\Delta_{\min} > 0$, we have distribution-dependent bound*

$$Reg_{\boldsymbol{\mu},\alpha,\beta}(T) \le \sum_{i\in[m]} \frac{48B^2 K \ln T}{\Delta_{\min}^i} + 2Bm + \frac{\pi^2}{3}\cdot m\cdot\Delta_{\max}; \tag{5}$$

2. *we have distribution-independent bound*

$$Reg_{\boldsymbol{\mu},\alpha,\beta}(T) \le 14B\sqrt{KmT\ln T} + 2Bm + \frac{\pi^2}{3}\cdot m\cdot\Delta_{\max}; \tag{6}$$

*Proof of Theorem 4.* To unify the proofs for distribution-dependent and distribution-independent bounds, we introduce a positive real number $M_i$ for each arm $i$. Let $\mathcal{F}_t$ be the event $\{r_{S_t}(\bar{\boldsymbol{\mu}}) < \alpha \cdot \mathrm{opt}(\bar{\boldsymbol{\mu}})\}$. In other words, $\mathcal{F}_t$ means the oracle fails in round $t$. By assumption, $\Pr\{\mathcal{F}_t\} \leq 1 - \beta$. Define $M_S = \max_{i \in \tilde{S}} M_i$ for each action $S$, specifically, $M_S = 0$ if $\tilde{S} = \varnothing$. Define

$$\kappa_T(M, s) = \begin{cases} 2B, & \text{if } s = 0, \\ 2B\sqrt{\frac{6 \ln T}{s}}, & \text{if } 1 \leq s \leq \ell_T(M), \\ 0, & \text{if } s \geq \ell_T(M) + 1, \end{cases}$$

where

$$\ell_T(M) = \left\lfloor \frac{24 B^2 K^2 \ln T}{M^2} \right\rfloor.$$

We then show that if $\{\Delta_{S_t} \geq M_{S_t}\}$, $\neg \mathcal{F}_t$ and $\mathcal{N}_t^{\mathrm{s}}$ hold, we have

$$\Delta_{S_t} \leq \sum_{i \in \tilde{S}_t} \kappa_T(M_i, T_{i,t-1}). \tag{7}$$

The right hand side of the inequality is non-negative, so it holds naturally if $\Delta_{S_t} = 0$. We only need to consider $\Delta_{S_t} > 0$. By $\mathcal{N}_t^{\mathrm{s}}$ and $\neg \mathcal{F}_t$, we have

$$r_{S_t}(\bar{\boldsymbol{\mu}}_t) \geq \alpha \cdot \mathrm{opt}(\bar{\boldsymbol{\mu}}_t) \geq \alpha \cdot \mathrm{opt}(\boldsymbol{\mu}) = r_{S_t}(\boldsymbol{\mu}) + \Delta_{S_t},$$

Then by Condition 2,

$$\Delta_{S_t} \leq r_{S_t}(\bar{\boldsymbol{\mu}}_t) - r_{S_t}(\boldsymbol{\mu}) \leq B \sum_{i \in \tilde{S}_t} (\bar{\mu}_{i,t} - \mu_i).$$

We are going to bound $\Delta_{S_t}$ by bounding $\bar{\mu}_{i,t} - \mu_i$. But before doing so, we first perform a transformation. As we have $\Delta_{S_t} \geq M_{S_t}$, so $B \sum_{i \in \tilde{S}_t}(\bar{\mu}_{i,t} - \mu_i) \geq \Delta_{S_t} \geq M_{S_t}$. We have

$$
\begin{aligned}
\Delta_{S_t} &\leq B \sum_{i \in \tilde{S}_t} (\bar{\mu}_{i,t} - \mu_i) \\
&\leq -M_{S_t} + 2B \sum_{i \in \tilde{S}_t} (\bar{\mu}_{i,t} - \mu_i) \\
&= 2B \sum_{i \in \tilde{S}_t} \left[ (\bar{\mu}_{i,t} - \mu_i) - \frac{M_{S_t}}{2B \left| \tilde{S}_t \right|} \right] \\
&\leq 2B \sum_{i \in \tilde{S}_t} \left[ (\bar{\mu}_{i,t} - \mu_i) - \frac{M_{S_t}}{2BK} \right] \\
&\leq 2B \sum_{i \in \tilde{S}_t} \left[ (\bar{\mu}_{i,t} - \mu_i) - \frac{M_i}{2BK} \right]. 
\end{aligned}
\tag{8}
$$

By $\mathcal{N}_t^{\mathrm{s}}$, we have $\bar{\mu}_{i,t} - \mu_i \leq \min\{2\rho_{i,t}, 1\}$. So

$$(\bar{\mu}_{i,t} - \mu_i) - \frac{M_i}{2BK} \leq \min\{2\rho_{i,t}, 1\} - \frac{M_i}{2BK} \leq \min\left\{2\sqrt{\frac{3 \ln T}{2 T_{i,t-1}}}, 1\right\} - \frac{M_i}{2BK}.$$

If $T_{i,t-1} \leq \ell_T(M_i)$, we have $(\bar{\mu}_{i,t} - \mu_i) - \frac{M_i}{2BK} \leq \min\left\{2\sqrt{\frac{3 \ln T}{2 T_{i,t-1}}}, 1\right\} \leq \frac{1}{2B}\kappa_T(M_i, T_{i,t-1})$. If $T_{i,t-1} \geq \ell_T(M_i) + 1$, then $2\sqrt{\frac{3 \ln T}{2 T_{i,t-1}}} \leq \frac{M_i}{2BK}$, so $(\bar{\mu}_{i,t} - \mu_i) - \frac{M_i}{2BK} \leq 0 = \frac{1}{2B}\kappa_T(M_i, T_{i,t-1})$. In conclusion, we continue (8) with

$$(8) \leq \sum_{i \in \tilde{S}_t} \kappa_T(M_i, T_{i,t-1}).$$

Then in each run,

$$\sum_{t=1}^{T} \mathbb{I}(\{\Delta_{S_t} \geq M_{S_t}\} \wedge \neg\mathcal{F}_t \wedge \mathcal{N}_t^s) \cdot \Delta_{S_t} \leq \sum_{t=1}^{T} \sum_{i \in \tilde{S}_t} \kappa_T(M_i, T_{i,t-1})$$

$$= \sum_{i \in [m]} \sum_{s=0}^{T_{i,T}} \kappa_T(M_i, s)$$

$$\leq \sum_{i \in [m]} \sum_{s=0}^{\ell_T(M_i)} \kappa_T(M_i, s)$$

$$= 2Bm + \sum_{i \in [m]} \sum_{s=1}^{\ell_T(M_i)} 2B\sqrt{\frac{6 \ln T}{s}}$$

$$\leq 2Bm + \sum_{i \in [m]} \int_{s=0}^{\ell_T(M_i)} 2B\sqrt{\frac{6 \ln T}{s}}\mathrm{d}s$$

$$\leq 2Bm + \sum_{i \in [m]} 4B\sqrt{6 \ln T \ell_T(M_i)}$$

$$\leq 2Bm + \sum_{i \in [m]} 4B\sqrt{6 \ln T \cdot \frac{24B^2 K^2 \ln T}{M_i^2}}$$

$$\leq 2Bm + \sum_{i \in [m]} \frac{48B^2 K \ln T}{M_i}.$$

So

$$Reg(\{\Delta_{S_t} \geq M_{S_t}\} \wedge \neg\mathcal{F}_t \wedge \mathcal{N}_t^s) = \mathbb{E}\left[\sum_{t=1}^{T} \mathbb{I}(\{\Delta_{S_t} \geq M_{S_t}\} \wedge \neg\mathcal{F}_t \wedge \mathcal{N}_t^s) \cdot \Delta_{S_t}\right]$$

$$\leq 2Bm + \sum_{i \in [m]} \frac{48B^2 K \ln T}{M_i}.$$

By Lemma 3, $\Pr\{\neg\mathcal{N}_t^s\} \leq 2mt^{-2}$. Then, as $Reg(\mathcal{E}_t) \leq \sum_{t=1}^{T} \Pr\{\mathcal{E}_t\}\Delta_{\max}$ by definition of filtered regret,

$$Reg(\neg\mathcal{N}_t^s) \leq \sum_{t=1}^{T} 2mt^{-2} \cdot \Delta_{\max} \leq \frac{\pi^2}{3}m \cdot \Delta_{\max},$$

$$Reg(\mathcal{F}_t) \leq (1-\beta)T \cdot \Delta_{\max}.$$

The filtered regret with null event

$$Reg(\{\}) \leq Reg(\neg\mathcal{N}_t^s) + Reg(\mathcal{F}_t) + Reg(\Delta_{S_t} < M_{S_t}) + Reg(\{\Delta_{S_t} \geq M_{S_t}\} \wedge \neg\mathcal{F}_t \wedge \mathcal{N}_t^s)$$

$$\leq (1-\beta)T \cdot \Delta_{\max} + \frac{\pi^2}{3}m \cdot \Delta_{\max} + 2Bm + \sum_{i \in [m]} \frac{48B^2 K \ln T}{M_i} + Reg(\Delta_{S_t} < M_{S_t}).$$

By definition of filtered regret, $Reg_{\boldsymbol{\mu},\alpha,\beta}(T) = Reg(T, \{\}) - (1-\beta)T \cdot \Delta_{\max}$, so

$$Reg_{\boldsymbol{\mu},\alpha,\beta}(T) \leq \frac{\pi^2}{3}m \cdot \Delta_{\max} + 2Bm + \sum_{i \in [m]} \frac{48B^2 K \ln T}{M_i} + Reg(\Delta_{S_t} < M_{S_t}).$$

For distribution-dependent bound, take $M_i = \Delta_{\min}^i$, then $Reg(\Delta_{S_t} < M_{S_t}) = 0$ and we have

$$Reg_{\boldsymbol{\mu},\alpha,\beta}(T) \leq \sum_{i \in [m]} \frac{48B^2 K \ln T}{M_i} + 2Bm + \frac{\pi^2}{3} \cdot \Delta_{\max}.$$

For distribution-independent bound, take $M_i = M = \sqrt{(48B^2mK\ln T)/T}$, then $Reg(\Delta_{S_t} < M_{S_t}) \leq TM$ and we have

$$
\begin{aligned}
Reg_{\boldsymbol{\mu},\alpha,\beta}(T) &\leq \sum_{i \in [m]} \frac{48B^2 K \ln T}{M_i} + 2Bm + \frac{\pi^2}{3} m \cdot \Delta_{\max} + Reg(\Delta_{S_t} < M_{S_t}) \\
&\leq \frac{48B^2 mK \ln T}{M} + 2Bm + \frac{\pi^2}{3} m \cdot \Delta_{\max} + TM \\
&= 2\sqrt{48B^2 mKT \ln T} + \frac{\pi^2}{3} m \cdot \Delta_{\max} + 2Bm \\
&\leq 14B\sqrt{mKT \ln T} + \frac{\pi^2}{3} m \cdot \Delta_{\max} + 2Bm. \qquad \square
\end{aligned}
$$

### B.3 Proof of Theorem 1 (1-Norm Case Regret Bound)

We first show the distribution-dependent upper bound (Eq. (1)) and the distribution-independent upper bound below, which is a weaker version of Eq. (2):

$$
Reg_{\boldsymbol{\mu},\alpha,\beta}(T) \leq 48B\sqrt{mKT \ln T} + \left( \left\lceil \log_2 \sqrt{\frac{KT}{288m \ln T}} \right\rceil_0 + 2 \right) \cdot m \cdot \frac{\pi^2}{6} \cdot \Delta_{\max} + 4Bm. \tag{9}
$$

We show full proof of Eq. (2) later in Section B.3.1. The proof of Eq. (9) is based on the distribution-dependent bound (Eq. (1)) similar to other analysis, and thus could be more familiar to readers and easier to follow, while Eq. (2) has better constant and requires an independent proof as given in Section B.3.1.

Let $\mathcal{F}_t$ be the event $\{r_{S_t}(\bar{\boldsymbol{\mu}}) < \alpha \cdot \mathrm{opt}(\bar{\boldsymbol{\mu}})\}$. In other words, $\mathcal{F}_t$ means the oracle fails in round $t$. By assumption, $\Pr\{\mathcal{F}_t\} \leq 1 - \beta$.

To unify the proofs for distribution-dependent and distribution-independent bounds, we introduce a positive real number $M_i$ for each arm $i$. Define $M_S = \max_{i \in \tilde{S}} M_i$ for each action $S$, specifically, $M_S = 0$ if $\tilde{S} = \varnothing$. To prove the distribution-dependent bound, we will let $M_i = \Delta_{\min}^i$. To prove the distribution-independent bound, we will let $M_i = M = \tilde{\Theta}(T^{-1/2})$ to balance bounds for $Reg(\{\Delta_{S_t} \geq M_{S_t}\})$ and $Reg(\{\Delta_{S_t} < M_{S_t}\})$. Implement definition of $\mathcal{N}_t^i$ (Definition 7) with $j_{\max}^i = j_{\max}(M_i) = \left\lceil \log_2 \frac{2BK}{M_i} \right\rceil_0$. Define

$$
\kappa_{j,T}(M, s) = \begin{cases} 4 \cdot 2^{-j} B, & \text{if } s = 0, \\ 2B\sqrt{\frac{72 \cdot 2^{-j} \ln T}{s}}, & \text{if } 1 \leq s \leq \ell_{j,T}(M), \\ 0, & \text{if } s \geq \ell_{j,T}(M) + 1, \end{cases}
$$

where

$$
\ell_{j,T}(M) = \left\lfloor \frac{288 \cdot 2^{-j} B^2 K^2 \ln T}{M^2} \right\rfloor,
$$

and the following lemma explains that $\kappa$ is the contribution to regret.

**Lemma 5.** *In every run of the CUCB algorithm on a problem instance that satisfies 1-norm TPM bounded smoothness (Condition 2), for any vector $\{M_i\}_{i \in [m]}$ of positive real numbers and $1 \leq t \leq T$, if $\{\Delta_{S_t} \geq M_{S_t}\}, \neg\mathcal{F}_t, \mathcal{N}_t^s$ and $\mathcal{N}_t^t$ hold, we have*

$$
\Delta_{S_t} \leq \sum_{i \in \tilde{S}_t} \kappa_{j_i, T}(M_i, N_{i, j_i, t-1}),
$$

*where $j_i$ is the index of the TP group with $S_t \in \mathcal{S}_{i, j_i}$ (See Definition 5).*

*Proof.* The right hand side of the inequality is non-negative, so it holds naturally if $\Delta_{S_t} = 0$. We only need to consider $\Delta_{S_t} > 0$. By $\mathcal{N}_t^s$ and $\neg\mathcal{F}_t$, we have

$$
r_{S_t}(\bar{\boldsymbol{\mu}}_t) \geq \alpha \cdot \mathrm{opt}(\bar{\boldsymbol{\mu}}_t) \geq \alpha \cdot \mathrm{opt}(\boldsymbol{\mu}) = r_{S_t}(\boldsymbol{\mu}) + \Delta_{S_t},
$$

Then by Condition 2,

$$\Delta_{S_t} \le r_{S_t}(\bar{\boldsymbol{\mu}}_t) - r_{S_t}(\boldsymbol{\mu}) \le B \sum_{i \in \tilde{S}_t} p_i^{D,S_t}(\bar{\mu}_{i,t} - \mu_i).. \tag{10}$$

We are going to bound $\Delta_{S_t}$ by bounding $p_i^{D,S_t}(\bar{\mu}_{i,t} - \mu_i)$. But before doing so, we first perform a transformation. As we have $\Delta_{S_t} \ge M_{S_t}$, so $B \sum_{i \in \tilde{S}_t} p_i^{D,S_t}(\bar{\mu}_{i,t} - \mu_i) \ge \Delta_{S_t} \ge M_{S_t}$. We have

$$\begin{aligned}
\Delta_{S_t} &\le B \sum_{i \in \tilde{S}_t} p_i^{D,S_t}(\bar{\mu}_{i,t} - \mu_i) \\
&\le -M_{S_t} + 2B \sum_{i \in \tilde{S}_t} p_i^{D,S_t}(\bar{\mu}_{i,t} - \mu_i) \\
&= 2B \sum_{i \in \tilde{S}_t} \left[ p_i^{D,S_t}(\bar{\mu}_{i,t} - \mu_i) - \frac{M_{S_t}}{2B \left| \tilde{S}_t \right|} \right] \\
&\le 2B \sum_{i \in \tilde{S}_t} \left[ p_i^{D,S_t}(\bar{\mu}_{i,t} - \mu_i) - \frac{M_{S_t}}{2BK} \right] \\
&\le 2B \sum_{i \in \tilde{S}_t} \left[ p_i^{D,S_t}(\bar{\mu}_{i,t} - \mu_i) - \frac{M_i}{2BK} \right]. \tag{11}
\end{aligned}$$

Then we bound $p_i^{D,S_t}(\bar{\mu}_{i,t} - \mu_i)$. By $\mathcal{N}_t^{\mathrm{s}}$,

$$\bar{\mu}_{i,t} - \mu_i < 2\rho_{i,t} = 2\sqrt{\frac{3\ln t}{2T_{i,t-1}}}.$$

Both $\bar{\mu}_{i,t}$ and $\mu_i$ are in $[0,1]$, so $\bar{\mu}_{i,t} - \mu_i \le 1$. We then bound $p_i^{D,S_t}(\bar{\mu}_{i,t} - \mu_i)$ in different cases.

- *Case I:* $1 \le j_i \le j_{\max}^i$. Then we have $p_i^{D,S_t} \le 2 \cdot 2^{-j_i}$. If $\sqrt{\frac{6\ln t}{\frac{1}{3}N_{i,j_i,t-1} \cdot 2^{-j_i}}} \le 1$, by $\mathcal{N}_t^{\mathrm{t}}$,

$$\bar{\mu}_{i,t} - \mu_i \le 2\sqrt{\frac{3\ln t}{2T_{i,t-1}}} \le \sqrt{\frac{6\ln t}{\frac{1}{3}N_{i,j_i,t-1} \cdot 2^{-j_i}}},$$

so

$$\bar{\mu}_{i,t} - \mu_i \le \min\left\{ \sqrt{\frac{6\ln t}{\frac{1}{3}N_{i,j_i,t-1} \cdot 2^{-j_i}}}, 1 \right\},$$

and

$$\begin{aligned}
&p_i^{D,S_t}(\bar{\mu}_{i,t} - \mu_i) \\
&\le 2 \cdot 2^{-j_i} \cdot \min\left\{ \sqrt{\frac{6\ln t}{\frac{1}{3}N_{i,j_i,t-1} \cdot 2^{-j_i}}}, 1 \right\} \\
&= \min\left\{ \sqrt{\frac{72 \cdot 2^{-j_i}\ln T}{N_{i,j_i,t-1}}}, 2 \cdot 2^{-j_i} \right\}.
\end{aligned}$$

If $N_{i,j_i,t-1} \ge \ell_{j_i,T}(M_i) + 1$, then $\sqrt{\frac{72 \cdot 2^{-j_i}\ln T}{N_{i,j_i,t-1}}} \le \frac{M_i}{2BK}$ and $p_i^{D,S_t}(\bar{\mu}_{i,t} - \mu_i) - \frac{M_i}{2BK} \le 0$. If $N_{i,j_i,t-1} = 0$, we use the bound $p_i^{D,S_t}(\bar{\mu}_{i,t} - \mu_i) \le 2 \cdot 2^{-j_i}$. Otherwise, i.e. $1 \le N_{i,j_i,t-1} \le \ell_{j_i,T}(M_i)$, we use $p_i^{D,S_t}(\bar{\mu}_{i,t} - \mu_i) \le \sqrt{\frac{72 \cdot 2^{-j_i}\ln T}{N_{i,j_i,t-1}}}$. Recall the definition of $\kappa_{j,T}(M,s)$, then, for $1 \le j_i \le j_{\max}^i$, we have

$$p_i^{D,S_t}(\bar{\mu}_{i,t} - \mu_i) - \frac{M_i}{2BK} \le \frac{1}{2B}\kappa_{j_i,T}(M_i, N_{i,j_i,t-1}). \tag{12}$$

- *Case II:* $j_i \geq j_{\max}^i + 1 = \left\lceil \log_2 \frac{2BK}{M_i} \right\rceil_0 + 1$. Then we have

$$p_i^{D,S_t}(\bar{\mu}_{i,t} - \mu_i) \leq p_i^{D,S_t} \leq 2 \cdot 2^{-j_i}$$

$$\leq 2 \cdot 2^{-\log_2 \frac{2BK}{M_i} - 1} = \frac{M_i}{2BK}.$$

So

$$p_i^{D,S_t}(\bar{\mu}_{i,t} - \mu_i) - \frac{M_i}{2BK} \leq 0 \leq \frac{1}{2B} \kappa_{j_i,T}(M_i, N_{i,j_i,t-1}). \tag{13}$$

Combining Eq. (11), (12) and (13), we conclude the proof with

$$\Delta_{S_t} \leq 2B \sum_{i \in \tilde{S}_t} \left[ p_i^{D,S_t}(\bar{\mu}_{i,t} - \mu_i) - \frac{M_i}{2BK} \right]$$

$$\leq \sum_{i \in \tilde{S}_t} \kappa_{j_i,T}(M_i, N_{i,j_i,t-1}). \qquad \square$$

We remark that the proof of Lemma 5, in particular the derivation leading to Eq. (11) together with the argument in the paragraph before Eq.(12), contains the reverse amortization trick we mentioned in the main text. In particular, by the derivation of Eq. (11), the contribution of every arm $i$ to regret $\Delta_{S_t}$ is accounted as $2B \left[ p_i^{D,S_t}(\bar{\mu}_{i,t} - \mu_i) - \frac{M_i}{2BK} \right]$. Then by the argument in the paragraph before Eq.(12), if $N_{i,j_i,t-1} \geq \ell_{j_i,T}(M_i) + 1$, meaning that $i$ has been triggered by actions in group $j_i$ for at least $\ell_{j_i,T}(M_i) + 1$, its error $|\bar{\mu}_{i,t} - \mu_i|$ would be small enough such that its contribution to the regret $\Delta_{S_t}$ is not positive. This trick eliminates the need of summing up small errors from many sufficiently sampled arms, leading to a tighter regret bound. The same trick can be seen in Appendix B.2, Eq.(8) and the derivation that follows for the no triggered arm case.

**Lemma 6.** *For the CUCB algorithm on a problem instance that satisfies TPM bounded smoothness with 1-norm (Condition 2),*

$$Reg(\{\Delta_{S_t} \geq M_{S_t}\} \wedge \neg \mathcal{F}_t \wedge \mathcal{N}_t^s \wedge \mathcal{N}_t^t) \leq \sum_{i \in [m]} \frac{576 B^2 K \ln T}{M_i} + 4Bm.$$

*Proof.* We bound $Reg(\{\Delta_{S_t} \geq M_{S_t}\} \wedge \neg \mathcal{F}_t \wedge \mathcal{N}_t^s \wedge \mathcal{N}_t^t)$ with Lemma 5. In every run,

$$\sum_{t=1}^{T} \mathbb{I}(\{\Delta_{S_t} \geq M_{S_t}\} \wedge \neg \mathcal{F}_t \wedge \mathcal{N}_t^s \wedge \mathcal{N}_t^t) \Delta_{S_t} \leq \sum_{t=1}^{T} \sum_{i \in \tilde{S}_t} \kappa_{j_i,T}(M_i, N_{i,j_i,t-1})$$

$$= \sum_{i \in [m]} \sum_{j=1}^{+\infty} \sum_{s=0}^{N_{i,j,T}-1} \kappa_{j,T}(M_i, s), \tag{14}$$

where (14) is due to $N_{i,j_i}$ is increased if and only if $i \in \tilde{S}_t$. For every arm $i$ and $j \geq 1$,

$$\sum_{s=0}^{N_{i,j,T}-1} \kappa_{j,T}(M_i, s) \leq \sum_{s=0}^{\ell_{j,T}(M_i)} \kappa_{j,T}(M_i, s) \tag{15}$$

$$= \kappa_{j,T}(M_i, 0) + \sum_{s=1}^{\ell_{j,T}(M_i)} \kappa_{j,T}(M_i, s)$$

$$= \kappa_{j,T}(M_i, 0) + \sum_{s=1}^{\ell_{j,T}(M_i)} 2B \sqrt{\frac{72 \cdot 2^{-j_i} \ln T}{s}}$$

$$\leq \kappa_{j,T}(M_i, 0) + 4B \sqrt{72 \cdot 2^{-j_i} \ln T} \sqrt{\ell_{j,T}(M_i)}, \tag{16}$$

where(15) is due to $\kappa_{j,T}(s) = 0$ when $s \geq \ell_{j,T}(M) + 1$, and (16) is due to the fact that, for every natural number integer $n$,

$$\sum_{s=1}^{n} \sqrt{\frac{1}{s}} \leq \int_{s=0}^{n} \sqrt{\frac{1}{s}} \, ds = 2\sqrt{n}.$$

By definition, $\ell_{j,T}(M_i) \leq \frac{288 \cdot 2^{-j_i} B^2 K^2 \ln T}{M_i^2}$, so

$$(16) \leq \kappa_{j,T}(M, 0) + 4B\sqrt{72 \cdot 2^{-j_i} \ln T}\sqrt{\frac{288 \cdot 2^{-j_i} B^2 K^2 \ln T}{M_i^2}}$$

$$= 4 \cdot 2^{-j} B + \frac{576 \cdot 2^{-j_i} B^2 K \ln T}{M_i}.$$

Then we continue (14) with

$$(14) \leq \sum_{i \in [m]} \sum_{j=1}^{+\infty} \left( 4 \cdot 2^{-j} B + \frac{576 \cdot 2^{-j_i} B^2 K \ln T}{M_i} \right)$$

$$= \sum_{i \in [m]} \left[ \left( 4B + \frac{576 B^2 K \ln T}{M_i} \right) \cdot \sum_{j=1}^{+\infty} 2^{-j} \right]$$

$$= \sum_{i \in [m]} \left( 4B + \frac{576 B^2 K \ln T}{M_i} \right)$$

$$= \sum_{i \in [m]} \frac{576 B^2 K \ln T}{M_i} + 4Bm.$$

By taking expectation over all possible runs,

$$Reg(\{\Delta_{S_t} \geq M_{S_t}\} \wedge \neg \mathcal{F}_t \wedge \mathcal{N}_t^{\mathrm{s}} \wedge \mathcal{N}_t^{\mathrm{t}}) = \mathbb{E}[\mathbb{I}(\{\Delta_{S_t} \geq M\} \wedge \neg \mathcal{F}_t \wedge \mathcal{N}_t^{\mathrm{s}} \wedge \mathcal{N}_t^{\mathrm{t}})\Delta_{S_t}]$$

$$\leq \sum_{i \in [m]} \frac{576 B^2 K \ln T}{M_i} + 4Bm. \qquad \square$$

*Proof of Theorem 1.* Recall Definition 3, the definition of event-filtered regret:

$$Reg_{\boldsymbol{\mu}}^A(T, \{\mathcal{E}_t\}_{t \geq 1}) = \mathbb{E}\left[\sum_{t=1}^T \mathbb{I}(\mathcal{E}_t)(\alpha \cdot \mathrm{opt}_{\boldsymbol{\mu}} - r_{S_t^A}(\boldsymbol{\mu}))\right] = T \cdot \alpha \cdot \mathrm{opt}_{\boldsymbol{\mu}} - \mathbb{E}\left[\sum_{t=1}^T \mathbb{I}(\mathcal{E}_t)(r_{S_t^A}(\boldsymbol{\mu}))\right].$$

Then for filtered regret with null event (the event that is always true), we have $Reg(\{\}) = Reg_{\boldsymbol{\mu}, \alpha, \beta} + (1 - \beta)T \cdot \alpha \cdot \mathrm{opt}_{\boldsymbol{\mu}}$. We divide this filtered regret into parts as

$$Reg(\{\}) \leq Reg(\{\Delta_{S_t} < M_{S_t}\}) + Reg(\mathcal{F}_t) + Reg(\neg \mathcal{N}_t^{\mathrm{s}}) + Reg(\neg \mathcal{N}_t^{\mathrm{t}})$$
$$+ Reg(\{\Delta_{S_t} \geq M_{S_t}\} \wedge \neg \mathcal{F}_t \wedge \mathcal{N}_t^{\mathrm{s}} \wedge \mathcal{N}_t^{\mathrm{t}}). \tag{17}$$

By definition of filtered regret, $Reg(\mathcal{E}_t) \leq \sum_{t=1}^T \mathbb{I}\{\mathcal{E}_t\}\Delta_{S_t} \leq \sum_{t=1}^T \Pr\{\mathcal{E}_t\} \cdot \Delta_{\max}$, then

$$Reg(\mathcal{F}_t) \leq \sum_{t=1}^T \Pr\{\mathcal{F}_t\}\Delta_{\max} = (1 - \beta)T \cdot \Delta_{\max}, \tag{18}$$

$$Reg(\neg \mathcal{N}_t^{\mathrm{s}}) \leq \sum_{t=1}^T \Pr\{\neg \mathcal{N}_t^{\mathrm{s}}\}\Delta_{\max} \leq \frac{\pi^2}{3} \cdot m \cdot \Delta_{\max}, \tag{19}$$

$$Reg(\neg \mathcal{N}_t^{\mathrm{t}}) \leq \sum_{t=1}^T \Pr\{\neg \mathcal{N}_t^{\mathrm{t}}\}\Delta_{\max} \leq \frac{\pi^2}{6} \cdot \sum_{i \in [m]} j_{\max}^i \cdot \Delta_{\max}. \tag{20}$$

By Lemma 6,

$$Reg(\{\Delta_{S_t} \geq M_{S_t}\} \wedge \neg \mathcal{F}_t \wedge \mathcal{N}_t^{\mathrm{s}} \wedge \mathcal{N}_t^{\mathrm{t}}) \leq \sum_{i \in [m]} \frac{576 B^2 K \ln T}{M_i} + 4Bm.$$

Take $M_i = \Delta^i_{\min}$. If $\Delta_{S_t} < M_{S_t}$, then $\Delta_{S_t} = 0$, since we have either $\tilde{S}_t = \varnothing$ or $\Delta_{S_t} < M_{S_t} \le M_i$ for some $i \in \tilde{S}_t$. So $Reg(\{\Delta_{S_t} < M_{S_t}\}) = 0$. Then we have

$$Reg(\{\}) \le (1-\beta)T \cdot \Delta_{\max} + \sum_{i \in [m]} \frac{576B^2 K \ln T}{\Delta^i_{\min}} + 4Bm + \frac{\pi^2}{6} \cdot \sum_{i \in [m]} \left(j_{\max}(\Delta^i_{\min}) + 2\right) \cdot \Delta_{\max},$$
(21)

where we abuse the notation of $j_{\max}(M) = \left\lceil \log_2 \frac{2BK}{M_i} \right\rceil_0$.

On the other hand, take $M_i = M = \sqrt{(576B^2 mK \ln T)/T}$, then $\Delta_{S_t}$ is also $M$ for every action $S_t$ that $\tilde{S}_t$ is non-empty. We bound $Reg(\{\Delta_{S_t} < M\})$ with

$$Reg(\{\Delta_{S_t} < M_{S_t}\}) = \sum_{t=1}^T \mathbb{I}\{\Delta_{S_t} < M_{S_t}\}\Delta_{S_t} \le \sum_{t=1}^T \mathbb{I}\{\Delta_{S_t} < M_{S_t}\}M \le TM.$$

So the filtered regret with null event is bounded by

$$
\begin{aligned}
Reg(\{\}) &\le (1-\beta)T \cdot \Delta_{\max} + \frac{576B^2 mK \ln T}{M} + 4Bm + TM + \frac{\pi^2}{6} \cdot (j_{\max}(M) + 2) \cdot m \cdot \Delta_{\max} \\
&= (1-\beta)T \cdot \Delta_{\max} + \frac{576B^2 mK \ln T}{\sqrt{(576B^2 mK \ln T)/T}} + 4Bm + T\sqrt{(576B^2 mK \ln T)/T} \\
&\quad + \frac{\pi^2}{6} \cdot (j_{\max}(M) + 2) \cdot m \cdot \Delta_{\max} \\
&\le (1-\beta)T \cdot \Delta_{\max} + 48B\sqrt{mKT \ln T} + 4Bm + \frac{\pi^2}{6} \cdot (j_{\max}(M) + 2) \cdot m \cdot \Delta_{\max}.
\end{aligned}
$$
(22)

Since $Reg_{\boldsymbol{\mu},\alpha,\beta} = Reg(\{\}) - (1-\beta)T \cdot \alpha \cdot \mathrm{opt}_{\boldsymbol{\mu}} \le Reg(\{\}) - (1-\beta)T \cdot \Delta_{\max}$, (21) implies (1) and (22) implies (9). $\qquad \square$

### B.3.1 Further improvement on distribution-independent upper bound

We now prove the tighter distribution-independent bound (Eq. (2)) without going through distribution-dependent bound. We start with

$$\Delta_{S_t} \le B \sum_{i \in \tilde{S}_t} p_i^{D,S_t}(\bar{\mu}_{i,t} - \mu_i) \le B \sum_{i \in \tilde{S}_t} p_i^{D,S_t} \min\left\{1, 2\sqrt{\frac{3\ln T}{2T_{i,t-1}}}\right\},$$
(10)

when events $\neg \mathcal{F}_t$ and $\mathcal{N}_t^s$ are true. Use $j_{\max} = \left\lceil \log_2 \frac{T}{18 \ln T} \right\rceil_0$ to define $\mathcal{N}_t^t$. When $\mathcal{N}_t^t$, $\sqrt{\frac{3\ln T}{2T_{i,t-1}}} \le \sqrt{\frac{18 \cdot 2^{-j_i} \ln T}{N_{i,j_i,t-1}}}$ if $j_i \le j_{\max}$ by definition of $\mathcal{N}_t^t$, then $p_i^{D,S_t} \min\left\{1, 2\sqrt{\frac{3\ln T}{2T_{i,t-1}}}\right\} \le \min\left\{2^{-j_i+1}, \sqrt{\frac{72 \cdot 2^{-j_i} \ln T}{N_{i,j_i,t-1}}}\right\}$ as $p_i^{D,S_t} \le 2^{-j_i+1}$. If $j_i > j_{\max}$, we still have $p_i^{D,S_t} \le 2^{-j_i+1}$. Because $N_{i,j_i,t-1} < T$, we have $2^{j_i+1} \ge \sqrt{\frac{72 \cdot 2^{-j_i} \ln T}{N_{i,j_i,t-1}}}$. The conclusion is

$$p_i^{D,S_t} \min\left\{1, 2\sqrt{\frac{3\ln T}{2T_{i,t-1}}}\right\} \le \min\left\{2^{-j_i+1}, \sqrt{\frac{72 \cdot 2^{-j_i} \ln T}{N_{i,j_i,t-1}}}\right\}$$
(23)

always holds, regardless $j \le j_{\max}$ or $j > j_{\max}$. So we define $\kappa$ as following in this proof:

$$\kappa_{j,T}(s) = \min\left\{2B \cdot 2^{-j}, B\sqrt{\frac{72 \cdot 2^{-j} \ln T}{s}}\right\}.$$

According to (10) and (23),

$$Reg(\neg \mathcal{F}_t \wedge \mathcal{N}_t^{\mathrm{s}} \wedge \mathcal{N}_t^{\mathrm{t}}) \leq \sum_{t=1}^{T} \mathbb{I}(\neg \mathcal{F}_t \wedge \mathcal{N}_t^{\mathrm{s}} \wedge \mathcal{N}_t^{\mathrm{t}}) \Delta_{S_t}$$

$$\leq \sum_{t=1}^{T} \sum_{i \in \tilde{S}_t} \kappa_{j_i,T}(N_{i,j_i,t-1})$$

$$= \sum_{i \in [m]} \sum_{j=1}^{+\infty} \sum_{s=0}^{N_{i,j,T}-1} \kappa_{j,T}(s). \tag{24}$$

In each round, there are at most $K$ of the counters $\{N_{i,j}\}_{i \in [m], j \in \mathbb{N}^+}$ are increased by 1, so $\sum_{i \in [m]} \sum_{j=1}^{+\infty} N_{i,j,T} \leq KT$. To maximize the right hand side of (24) is to choose $KT$ largest elements from the multiset $\{\kappa_{j,T}(s)\}_{i \in [m], j \in \mathbb{N}^+, s \in \mathbb{N}}$, consider the continuous version below which is more tractable than finding $KT$ largest elements:

$$\sum_{i \in [m]} \sum_{j=1}^{+\infty} \sum_{s=0}^{N_{i,j,T}-1} \kappa_{j,T}(s) \leq \sum_{i \in [m]} \sum_{j=1}^{+\infty} \left( \kappa_{j,T}(0) + \sum_{s=1}^{\max\{0, N_{i,j,T}-1\}} \kappa_{j,T}(s) \right)$$

$$\leq 2Bm + \sum_{i \in [m]} \sum_{j=1}^{+\infty} \int_{s=0}^{N_{i,j,T}} \kappa_{j,T}(s) \mathrm{d}s$$

$$\leq 2Bm + \max_{\sum_{i,j} x_{i,j} \leq KT} \left[ \sum_{i \in [m]} \sum_{j=1}^{+\infty} \int_{s=0}^{x_{i,j}} B\sqrt{\frac{72 \cdot 2^{-j} \ln T}{s}} \mathrm{d}s \right]. \tag{25}$$

To maximize the above sum of integral, we must have $B\sqrt{\frac{72 \cdot 2^{-j} \ln T}{x_{i,j}}} = B\sqrt{\frac{72 \cdot 2^{-j'} \ln T}{x_{i',j'}}}$ for every $i, i' \in m, j, j' \in \mathbb{N}^+$. The solution is $x_{i,j} = 2^{-j} KT/m$. By taking the solution into (25), we have

$$(25) = 2Bm + \sum_{i \in [m]} \sum_{j=1}^{+\infty} \int_{s=0}^{2^{-j} KT/m} B\sqrt{\frac{72 \cdot 2^{-j} \ln T}{s}} \mathrm{d}s$$

$$= 2Bm + \sum_{i \in [m]} \sum_{j=1}^{+\infty} B\sqrt{144 \cdot 2^{-j} \cdot 2^{-j} KT \ln T/m}$$

$$= 2Bm + 12B\sqrt{mKT \ln T}. \tag{26}$$

Combining with Lemmas 3 & 4, we have

$$Reg(\{\}) \leq (1-\beta)T \cdot \Delta_{\max} + 12B\sqrt{mKT \ln T} + \left( \left\lceil \log_2 \frac{T}{18 \ln T} \right\rceil_0 + 2 \right) \cdot m \cdot \frac{\pi^2}{6} \cdot \Delta_{\max} + 2Bm,$$

implying (2).

### B.4 Refining Parameter $B$

We can refine 1-norm bounded smoothness (Condition 3) by replacing the parameter $B$ with a separate parameter $B_i$ for each arm $i$.

**Condition 4 (Refined 1-Norm TPM Bounded Smoothness).** *We say that a CMAB-T problem instance satisfies refined 1-norm TPM bounded smoothness, if there exists $B_i \in \mathbb{R}^+$ for every arm $i$ (referred as the* bounded smoothness constant*) such that, for any two distributions $D, D' \in \mathcal{D}$ with expectation vectors $\boldsymbol{\mu}$ and $\boldsymbol{\mu}'$, and any action $S$, we have $|r_S(\boldsymbol{\mu}) - r_S(\boldsymbol{\mu}')| \leq \sum_{i \in [m]} B_i p_i^{D,S} |\mu_i - \mu_i'|$.*

Then in Theorem 1, we may replace $B$ with $B_i$ in distribution-dependent bound and replace $B\sqrt{m}$ with $\sqrt{\sum_{i \in [m]} B_i^2}$ in distribution-independent bound, except that for the last constant term we replace

$Bm$ with $\sum_{i\in[m]} B_i$. More specifically, we have (1) if $\Delta_{\min} > 0$, we have distribution-dependent bound

$$Reg_{\boldsymbol{\mu},\alpha,\beta}(T) \leq \sum_{i\in[m]} \frac{576 B_i^2 K \ln T}{\Delta_{\min}^i} + \sum_{i\in[m]} \left( \left\lceil \log_2 \frac{2 B_i K}{\Delta_{\min}^i} \right\rceil_0 + 2 \right) \cdot \frac{\pi^2}{6} \cdot \Delta_{\max} + 4 \sum_{i\in[m]} B_i;$$
(27)

(2) we have distribution-independent bound

$$Reg_{\boldsymbol{\mu},\alpha,\beta}(T) \leq 12 \sqrt{\sum_{i\in[m]} B_i^2 K T \ln T} + \left( \left\lceil \log_2 \frac{T}{18 \ln T} \right\rceil_0 + 2 \right) \cdot m \cdot \frac{\pi^2}{6} \cdot \Delta_{\max} + 2 \sum_{i\in[m]} B_i.$$
(28)

The proof of this refinement is almost straightforward replacement of $B$ with $B_i$, except a few points that we want to highlight. The definition of $\kappa$ and $\ell$ will be

$$\kappa_{i,j,T}(M,s) = \begin{cases} 4 \cdot 2^{-j} B_i, & \text{if } s = 0, \\ 2 B_i \sqrt{\frac{72 \cdot 2^{-j} \ln T}{s}}, & \text{if } 1 \leq s \leq \ell_{i,j,T}(M), \\ 0, & \text{if } s \geq \ell_{i,j,T}(M) + 1, \end{cases}$$

where

$$\ell_{i,j,T}(M) = \left\lfloor \frac{288 \cdot 2^{-j} B_i^2 K^2 \ln T}{M^2} \right\rfloor.$$

To maximize the sum of integral in (25) (with $B$ replaced by $B_i$), we need $B_i \sqrt{\frac{72 \cdot 2^{-j} \ln T}{x_{i,j}}} = B_{i'} \sqrt{\frac{72 \cdot 2^{-j'} \ln T}{x_{i',j'}}}$ for every $i, i' \in [m]$ and $j, j' \in \mathbb{N}^+$. So $x_{i,j} \propto 2^{-j} B_i^2$, and then $x_{i,j} = 2^{-j} B_i^2 K T / \sum_{i\in[m]} B_i^2$.

## C  Proofs for Applications of CMAB-T (Lemmas 1 and 2 in Section 4.2)

### C.1  Proof of Lemma 1

*Proof.* Let $S$ be an action. We regard $S$ as a permutation of $k$ of the arms. Without loss of generality, we may assume $S = (1, \ldots, k)$ for some $k \leq K$. For convenience, we use $p_i^{\boldsymbol{\mu},S}$ instead of $p_i^{D,S}$, as arms are independent Bernoulli variables so that $D$ can be determined by $\boldsymbol{\mu}$. For an arm $i > k$, $i$ will not be triggered by action $S$, and thus $p_i^{\boldsymbol{\mu},S} = 0$. The reward also does not depend on those arms. So we may only consider the arms $1, \ldots, k$. For convenience, we only list the expectations of arms in $S$, so that $\boldsymbol{\mu} = (\mu_1, \ldots, \mu_k)$ and $\boldsymbol{\mu}' = (\mu_1', \ldots, \mu_k')$.

Informally speaking, we can change the expectation of the arms from $\mu_i$ to $\mu_i'$, in the reverse order from $k$ to 1. Changing the expectation of an arm $j$ does not affect the triggering probability of an arm $i$ ordered in front of $j$, i.e. $i < j$. And when changing an arm from $\mu_i$ to $\mu_i'$, the reward changes by at most $p_i^{\boldsymbol{\mu},S} |\mu_i - \mu_i'|$. Therefore the total difference of reward is at most $\sum_{i=1}^{k} p_i^{\boldsymbol{\mu},S} |\mu_i - \mu_i'|$.

Formally, for the conjunctive cascading bandit, $r_S(\boldsymbol{\mu}) = \prod_{j=1}^{k} \mu_j$, and $p_i^{\boldsymbol{\mu},S} = \prod_{j=1}^{i-1} \mu_j$ for $i = 1, \ldots, k$. For every $j = 0, 1, \ldots, k$, let

$$\boldsymbol{\mu}^{(j)} = (\mu_1, \ldots, \mu_j, \mu_{j+1}', \ldots, \mu_k'),$$

specifically, $\boldsymbol{\mu}^{(k)} = \boldsymbol{\mu}$, $\boldsymbol{\mu}^{(0)} = \boldsymbol{\mu}'$. Then,

$$
\begin{aligned}
\left| r_S(\boldsymbol{\mu}^{(j)}) - r_S(\boldsymbol{\mu}^{(j-1)}) \right| &= \left| \prod_{i=1}^{k} \mu_i^{(j)} - \prod_{i=1}^{k} \mu_i^{(j-1)} \right| \\
&= \prod_{i,i\neq j} \mu_i^{(j)} \left| \mu_j^{(j)} - \mu_j^{(j-1)} \right| \\
&\leq \prod_{i=1}^{j-1} \mu_i^{(j)} \left| \mu_j^{(j)} - \mu_j^{(j-1)} \right| \\
&= \prod_{i=1}^{j-1} \mu_i \left| \mu_j - \mu_j' \right| \\
&= p_j^{\boldsymbol{\mu},S} \left| \mu_j - \mu_j' \right|,
\end{aligned}
$$

$$
\begin{aligned}
|r_S(\boldsymbol{\mu}) - r_S(\boldsymbol{\mu}')| &= \left| r_S(\boldsymbol{\mu}^{(k)}) - r_S(\boldsymbol{\mu}^{(0)}) \right| \\
&\leq \sum_{j=1}^{k} \left| r_S(\boldsymbol{\mu}^{(j)}) - r_S(\boldsymbol{\mu}^{(j-1)}) \right| \\
&\leq \sum_{j=1}^{k} p_j^{\boldsymbol{\mu},S} \left| \mu_j - \mu_j' \right|.
\end{aligned}
$$

For the disjunctive case, let $\lambda_i = 1 - \mu_i$ for $i \in [m]$. Then we have $r_S(\boldsymbol{\mu}) = 1 - \prod_{j=1}^{k} \lambda_i$, and $p_i^{\boldsymbol{\mu},S} = \prod_{j=1}^{i-1} \lambda_j$. The rest analysis follows the same pattern as the conjunctive case. $\qquad\square$

### C.2  Proof of Lemma 2

#### C.2.1  Sufficient Condition

In influence maximization, there is a directed graph $G = (V, E)$. For convenience, we use an edge $e$ as the index, e.g. $\mu_e$. In this application, action $S$ is a set of at most $k$ nodes, so we also interpret $S$ as a set of nodes.

Recall TPM bounded smoothness (Condition 2). The formula that we need to satisfy is

$$
|r_S(\boldsymbol{\mu}) - r_S(\boldsymbol{\mu}')| \leq B \sum_{e \in E} p_e^{\boldsymbol{\mu},S} |\mu_e - \mu_e'|, \tag{29}
$$

where $B = \max_{u \in V} |\{v \in V \mid v \text{ can be reached from } u\}|$ for influence maximization bandit, and $p_e^{\boldsymbol{\mu},S}$ stands for $p_e^{D,S}$ as $D$ can be uniquely determined by $\boldsymbol{\mu}$.

Let $r_S^v(\boldsymbol{\mu})$ be the probability that $v$ is activated. We claim that if for every node $v$ and every $\boldsymbol{\mu}$ and $\boldsymbol{\mu}'$ vectors, we have

$$
|r_S^v(\boldsymbol{\mu}) - r_S^v(\boldsymbol{\mu}')| \leq \sum_{e \in E} p_e^{\boldsymbol{\mu},S} |\mu_e - \mu_e'|, \tag{30}
$$

Then we have Inequality (29). The reason is as follows. First, we show that Inequality (30) holds for all $\boldsymbol{\mu}$ and $\boldsymbol{\mu}'$ is equivalent to $|r_S^v(\boldsymbol{\mu}) - r_S^v(\boldsymbol{\mu}')| \leq \sum_{e \in E, e \text{ can reach } v} p_e^{\boldsymbol{\mu},S} |\mu_e - \mu_e'|$ for all $\boldsymbol{\mu}$ and $\boldsymbol{\mu}'$. In fact, the direction from the above inequality to Inequality (30) is trivial. For the reverse direction, let $\boldsymbol{\mu}''$ be an expectation vector such that for every edge $e$ that can reach $v$, $\mu_e'' = \mu_e'$, and for every edge $e$ that cannot reach $v$, $\mu_e'' = \mu_e$. Since the $r_S^v(\boldsymbol{\mu}')$ is only affected by edges that can reach $v$, we have $r_S^v(\boldsymbol{\mu}') = r_S^v(\boldsymbol{\mu}'')$. Then, we have $|r_S^v(\boldsymbol{\mu}) - r_S^v(\boldsymbol{\mu}')| = |r_S^v(\boldsymbol{\mu}) - r_S^v(\boldsymbol{\mu}'')| \leq \sum_{e \in E} p_e^{\boldsymbol{\mu},S} |\mu_e - \mu_e''| = \sum_{e \in E, e \text{ can reach } v} p_e^{\boldsymbol{\mu},S} |\mu_e - \mu_e'|$. Next, assuming $|r_S^v(\boldsymbol{\mu}) - r_S^v(\boldsymbol{\mu}')| \leq \sum_{e \in E, e \text{ can reach } v} p_e^{\boldsymbol{\mu},S} |\mu_e - \mu_e'|$

holds for all $v \in V$, we have

$$
\begin{aligned}
|r_S(\boldsymbol{\mu}) - r_S(\boldsymbol{\mu}')| &= |\sum_{v \in V} r_S^v(\boldsymbol{\mu}) - \sum_{v \in \Gamma(S)} r_S^v(\boldsymbol{\mu}')| \\
&\leq \sum_{v \in V} |r_S^v(\boldsymbol{\mu}) - r_S^v(\boldsymbol{\mu}')| \\
&\leq \sum_{v \in V} \sum_{e \in E, e \text{ can reach } v} p_e^{\boldsymbol{\mu}, S} |\mu_e - \mu_e'| \\
&= \sum_{e \in E} \sum_{v \in V, v \text{ can be reached from } e} p_e^{\boldsymbol{\mu}, S} |\mu_e - \mu_e'| \\
&\leq B \sum_{e \in E} p_e^{\boldsymbol{\mu}, S} |\mu_e - \mu_e'|.
\end{aligned}
$$

Thus, Inequality (29) holds.

Furthermore, we argue that it is sufficient to show that Inequality (30) holds when (1) $\boldsymbol{\mu} \leq \boldsymbol{\mu}'$, i.e. for every edge $e$, $\mu_e \leq \mu_e'$,; and (2) $|S| = 1$. The first condition is a straightforward conclusion from the Monotonicity condition (Condition 1). For the second condition, we may assume the seed set $S$ consists of only one node without loss of generality. Otherwise, we may add a super seed node $s^\circ$ and add edges from $s^\circ$ to $s$ and let $\mu_{(s^\circ, s)} = \mu'_{(s^\circ, s)} = 1$ for every node $s$ in $S$.

Therefore, in the rest of the proof of Lemma 2, we prove that the influence maximization bandit satisfies Inequality (30) for $\boldsymbol{\mu} \leq \boldsymbol{\mu}'$ and $|S| = 1$. Let $s$ be the single seed node, and $S = \{s\}$.

### C.2.2 Paths

In this subsection, we define an order of paths and assign the influence to the smallest path. Consider all the paths from $s$ to $v$. A path $L$ from $s$ to $v$ is a sequence of edges $(e_1 = (s, u_1), e_2 = (u_1, u_2), \ldots, e_{|L|} = (u_{|L|-1}, v))$. A simple path is a path that $s, v, u_1, \ldots, u_{|L|-1}$ are distinct.

We call each possible value of random vector $X$ an outcome and denote it with vector $\boldsymbol{x} \in \{0, 1\}^m$. We say an edge $e$ is *live* (with respect to $\boldsymbol{x}$) if the corresponding component of $\boldsymbol{x}$ is 1, i.e. influence can propagate through $e$ with the propagation under $\boldsymbol{x}$. Thus, connecting with the terminology in the influence maximization literature [12, 5], $\boldsymbol{x}$ corresponds to a *live-edge graph* in $G$, while $X$ corresponds to a *random live-edge graph*. We say a path $L$ is *live* (with respect to $\boldsymbol{x}$) if every edge of $L$ is live. Then we have $r_S^v(\boldsymbol{\mu}) = \Pr_{\boldsymbol{x} \sim X}\{\text{there is a live path from } s \text{ to } v \text{ in } \boldsymbol{x}\}$. For each $\boldsymbol{x}$ that contains a live path from $s$ to $v$, we designate a path to $\boldsymbol{x}$ as follows. We first list all the edges in an arbitrary order, and for every different edges $e_1$ and $e_2$, define $e_1 < e_2$ if $e_1$ appears before $e_2$. To compare two paths $L$ and $L'$, we first order the edges in $L$ and $L'$ in the descending order, respectively, and then compare them in the lexicographical order. In other words, to compare two paths, first compare their largest edges, if there is a tie, compare their second largest edges, and so on. If two paths continue to tie on edges and then one path ends with no more edges, then the shorter path is smaller. For every outcome $\boldsymbol{x}$ such that there is a live path from $s$ to $v$, we designate the smallest live path $L$ from $s$ to $v$ in $\boldsymbol{x}$ to $\boldsymbol{x}$. Then each path from $s$ to $v$ in the original graph $G$ has a subset of outcome $\boldsymbol{x}$'s that are designated to $L$, which means all paths from $s$ to $v$ partition all outcomes $\boldsymbol{x}$ by which path $\boldsymbol{x}$ is designated to. Thus, let $r_{v|L}^{\boldsymbol{\mu}, S} = \sum_{\boldsymbol{x} \text{ is designated to } L} \Pr[X = \boldsymbol{x}]$, namely the contribution of path $L$ through the outcome $\boldsymbol{x}$ designated to $L$, and we have $r_S^v(\boldsymbol{\mu}) = \sum_{L \text{ is a path from } s \text{ to } v} r_{v|L}^{\boldsymbol{\mu}, S}$. That is, we decompose $r_S^v(\boldsymbol{\mu})$ by $r_{v|L}^{\boldsymbol{\mu}, S}$'s according to paths $L$ from $s$ to $v$.

Before going further, we first figure out some basic properties of the smallest live path. The smallest live path must be simple, otherwise we can remove loops to get a smaller live path. Moreover, each substring of the smallest live path in $\boldsymbol{x}$ must also be the smallest in $\boldsymbol{x}$ for its respective starting and ending nodes. For a path $L = (e_1 = (u_0, u_1), e_2 = (u_1, u_2), \ldots, e_{|L|} = (u_{|L|-1}, u_{|L|}))$, a substring is a consecutive subsequence $L_1 = (e_i, e_{i+1}, \ldots, e_j)$. If $L$ is the smallest live path from $s$ to $v$ in $\boldsymbol{x}$, any substring $L_1$ must also be the smallest live path from $u$ to $w$ in $\boldsymbol{x}$, where $u$ and $w$ are the start and the end of $L_1$, respectively. Otherwise, if $L_2$ is a live path from $u$ to $w$ that smaller than $L_1$, then we can replace $L_1$ with $L_2$ in $L$ to get a smaller live path.

(a) A sample network        (b) Search tree with Node marked in each node

Figure 1: A sample network and its search tree

### C.2.3 Bypass

In this subsection, we define bypass, which is a tool for calculating the probability that a path is *not* the smallest. For a path $L = (e_1 = (u_0, u_1), e_2 = (u_1, u_2), \ldots, e_{|L|} = (u_{|L|-1}, u_{|L|}))$, a bypass is a path from $u_i$ to $u_j$ that

    (1) shares no edges with $L$;
    (2) is smaller than the substring of $L$ between $u_i$ and $u_j$.

A bypass is live (with respect to $\boldsymbol{x}$) is defined in the same way as a path being live. For a live path $L$ in $\boldsymbol{x}$ from some node $u_0$ to some other node $u_{|L|}$, if there is a live bypass of $L$, then $L$ cannot be the smallest live path from $u_0$ to $u_{|L|}$. The reverse also holds: if a live path $L$ has no live bypasses, then $L$ is the smallest live path from $u_0$ to $u_{|L|}$. To prove the reverse direction, assume that there is a live path $L'$ from $u_0$ to $u_{|L|}$ smaller than $L$. Let $e_i$ be the largest edge in $L$ that is not in $L'$. Because $L' < L$, such $e_i$ must exist, and moreover $e_i$ must be larger than all edges in $L'$ but not $L$. By breaking $L$ at $e_i$, we divide the nodes covered by $L$ into two parts, the start part and the end part. Let $w$ be the first node in $L'$ that is in the end part of $L$. Such node $w$ must exist because the end node $u_{|L|}$ is in the end part of $L$. Let $u$ be the last node in $L'$ that appears before $w$ in $L'$ and is in the start part of $L$. Such node $u$ must exist because the starting node $u_0$ is in the start part. Then the substring of $L'$ between $u$ and $w$ must share no edges with $L$. Otherwise, if the substring of $L'$ between $u$ and $w$ shares one edge $(u_j, u_{j+1})$ with $L$, $(u_j, u_{j+1})$ cannot be $e_i$, so $u$ cannot be $u_j$ and $w$ cannot be $u_{j+1}$. Then, (a) if $u_{j+1}$ is in the end part of $L$, then $u_{j+1}$ appearing before $w$ in $L'$ contradicts to $w$'s definition; and (b) if $u_{j+1}$ is in the start part of $L$, $u_{j+1}$ appearing after $u$ and before $w$ in $L'$ contradicts to the definition of $u$. Therefore, the substring of $L'$ between $u$ and $w$ shares no edges with $L$. Then since $e_i$ is larger than any edge in $L'$ and not in $L$, the substring of $L'$ between $u$ and $w$ is indeed a bypass of $L$.

For a path $L = (e_1 = (u_0, u_1), e_2 = (u_1, u_2), \ldots, e_{|L|} = (u_{|L|-1}, u_{|L|}))$, let $p_L^{\boldsymbol{\mu}, S}$ be the probability that $L$ is the smallest live path from its start to its end. Note that if $L$ is a path from $s$ to $v$, then we have $p_L^{\boldsymbol{\mu}, S} = r_{v|L}^{\boldsymbol{\mu}, S}$. With bypass, we have $p_L^{\boldsymbol{\mu}, S} = p_{1,L}^{\boldsymbol{\mu}, S} p_{2,L}^{\boldsymbol{\mu}, S}$, where $p_{1,L}^{\boldsymbol{\mu}, S}$ is the probability that $L$ is live and $p_{2,L}^{\boldsymbol{\mu}, S}$ is the probability that there is no live bypasses of $L$. It is clear that $p_{1,L}^{\boldsymbol{\mu}, S} = \prod_{i=1}^{|L|} \mu_{e_i}$, and $p_{2,L}^{\boldsymbol{\mu}, S}$ is the probability that some subset of edges in $E \setminus L$ forming a live bypass of $L$ does not occur. These two events are independent, since they are about two disjoint subsets of $E$.

### C.2.4 Bottom-up modification

We now describe the search tree formed from all simple paths from $s$ to $v$. We use $y, z$ to denote nodes in this tree. Each node $y$ is corresponding to a prefix of a path from $s$ to $v$, which is also a path denoted by $\mathrm{Path}(y)$. Denote the end node of $\mathrm{Path}(y)$ with $\mathrm{Node}(y)$. Denote the last edge of $\mathrm{Path}(y)$ with $\mathrm{Edge}(y)$. Denote the root of the tree with $\mathrm{root}$. $\mathrm{Path}(\mathrm{root})$ is the empty path $\varnothing$. Specifically, $\mathrm{Node}(\mathrm{root}) = s$, as $s$ is the start node of every path in our consideration. $\mathrm{Edge}(\mathrm{root})$ is undefined. For every non-root node $y$ in the tree, its parent is the node $z$ such that $\mathrm{Path}(z)$ is the $(|\mathrm{Path}(y)| - 1)$-prefix of $\mathrm{Path}(y)$. Figure 1 shows a sample of this tree structure.

For a node $y$ in the tree, we simplify the notation $p_{\text{Path}(y)}^{\boldsymbol{\mu},S}$ to $p_y^{\boldsymbol{\mu},S}$. Similarly, for a leaf node $y$ in the tree, we simplify the notation $r_{v|\text{Path}(y)}^{\boldsymbol{\mu},S}$ to $r_{v|y}^{\boldsymbol{\mu},S}$. Then we have $r_S^v(\boldsymbol{\mu}) = \sum_{y \text{ is leaf}} r_{v|y}^{\boldsymbol{\mu},S} = \sum_{y \text{ is a leaf}} p_y^{\boldsymbol{\mu},S}$.

We want to show that for all $\boldsymbol{\mu} \leq \boldsymbol{\mu}'$, we have

$$r_S^v(\boldsymbol{\mu}') - r_S^v(\boldsymbol{\mu}) = \sum_{y \text{ is a leaf}} \left( p_y^{\boldsymbol{\mu}',S} - p_y^{\boldsymbol{\mu},S} \right) \leq p_e^{\boldsymbol{\mu},S} \sum_{e \in E} (\mu_e' - \mu_e), \tag{31}$$

which is the same as Inequality (30) that we want to show.

Let $\boldsymbol{\mu}^{(y)}$ be the vector that

$$\mu_e^{(y)} = \begin{cases} \mu_e, & \text{if } e \in \text{Path}(y), \\ \mu_e', & \text{if } e \notin \text{Path}(y). \end{cases}$$

Thus we have $p_y^{\boldsymbol{\mu}^{(y)},S} = p_{1,y}^{\boldsymbol{\mu},S} p_{2,y}^{\boldsymbol{\mu}',S}$. Since for all edges $e \notin \text{Path}(y)$, $\mu_e \leq \mu_e'$, the probability that there is no live bypasses of $\text{Path}(y)$ is higher under $\boldsymbol{\mu}$ than under $\boldsymbol{\mu}'$, that is, $p_{2,y}^{\boldsymbol{\mu}',S} \leq p_{2,y}^{\boldsymbol{\mu},S}$. Therefore, $p_y^{\boldsymbol{\mu}^{(y)},S} \leq p_y^{\boldsymbol{\mu},S}$, which means that, to prove Inequality (31), it is enough to prove

$$\sum_{y \text{ is a leaf}} \left( p_y^{\boldsymbol{\mu}',S} - p_y^{\boldsymbol{\mu}^{(y)},S} \right) \leq p_e^{\boldsymbol{\mu},S} \sum_{e \in E} (\mu_e' - \mu_e). \tag{32}$$

We now consider the bottom-up modification of the expectations in $\text{Path}(y)$.

$$p_y^{\boldsymbol{\mu}',S} - p_y^{\boldsymbol{\mu}^{(y)},S} = \sum_{i=1}^{|\text{Path}(y)|} \left( p_y^{\boldsymbol{\mu}^{(z_{i-1})},S} - p_y^{\boldsymbol{\mu}^{(z_i)},S} \right), \tag{33}$$

where $z_i$ is the ancestor of $y$ at depth $i$. (Root has depth 0.) By switching summations and regrouping the summands $\left( p_y^{\boldsymbol{\mu}^{(z_{i-1})},S} - p_y^{\boldsymbol{\mu}^{(z_i)},S} \right)$ under $z_i$, we have

$$\sum_{y \text{ is a leaf}} \left( p_y^{\boldsymbol{\mu}',S} - p_y^{\boldsymbol{\mu}^{(y)},S} \right) = \sum_{y \text{ is a non-root node}} \sum_{z \text{ is a leaf under } y} \left( p_z^{\boldsymbol{\mu}^{(\text{Parent}(y))},S} - p_z^{\boldsymbol{\mu}^{(y)},S} \right). \tag{34}$$

We generalize the definition of $r_{v|y}^{\boldsymbol{\mu},S}$ to non-leaf nodes $y$ by

$$r_{v|y}^{\boldsymbol{\mu},S} = \sum_{z \text{ is a leaf under } y} p_z^{\boldsymbol{\mu},S}.$$

It is clear that this definition coincides the old one when $y$ is a leaf. Now

$$(34) = \sum_{y \text{ is a non-root node}} \left( r_{v|y}^{\boldsymbol{\mu}^{(\text{Parent}(y))},S} - r_{v|y}^{\boldsymbol{\mu}^{(y)},S} \right). \tag{35}$$

$$r_{v|y}^{\boldsymbol{\mu},S} = \sum_{z \text{ is a leaf under } y} p_z^{\boldsymbol{\mu},S} = \sum_{z \text{ is a leaf under } y} p_{1,z}^{\boldsymbol{\mu},S} p_{2,z}^{\boldsymbol{\mu},S} = p_{1,y}^{\boldsymbol{\mu},S} \sum_{z \text{ is a leaf under } y} \frac{p_{1,z}^{\boldsymbol{\mu},S}}{p_{1,y}^{\boldsymbol{\mu},S}} p_{2,z}^{\boldsymbol{\mu},S}.$$

$\frac{p_{1,z}^{\boldsymbol{\mu},S}}{p_{1,y}^{\boldsymbol{\mu},S}} p_{2,z}^{\boldsymbol{\mu},S}$ does not depend on $\mu_e$ for every $e \in \text{Path}(y)$. So

$$r_{v|y}^{\boldsymbol{\mu}^{(\text{Parent}(y))},S} - r_{v|y}^{\boldsymbol{\mu}^{(y)},S} = \left( p_{1,y}^{\boldsymbol{\mu}^{(\text{Parent}(y))},S} - p_{1,y}^{\boldsymbol{\mu}^{(y)},S} \right) \sum_{z \text{ is a leaf under } y} \frac{p_{1,z}^{\boldsymbol{\mu}',S}}{p_{1,y}^{\boldsymbol{\mu}',S}} p_{2,z}^{\boldsymbol{\mu}',S}$$

$$= \left( \mu_{\text{Edge}(y)}' - \mu_{\text{Edge}(y)} \right) p_{1,\text{Parent}(y)}^{\boldsymbol{\mu},S} \sum_{z \text{ is a leaf under } y} \frac{p_{1,z}^{\boldsymbol{\mu}',S}}{p_{1,y}^{\boldsymbol{\mu}',S}} p_{2,z}^{\boldsymbol{\mu}',S}. \tag{36}$$

| Topology | Bound in [25] | Our bound |
| --- | --- | --- |
| bar graphs | $\tilde{O}\left(\|V\|\sqrt{kT}\right)$ | $\tilde{O}\left(\sqrt{k\|V\|T}\right)$ |
| star graphs | $\tilde{O}\left(\|V\|^2\sqrt{kT}\right)$ | $\tilde{O}\left(\|V\|^2\sqrt{T}\right)$ |
| ray graphs | $\tilde{O}\left(\|V\|^{\frac{9}{4}}\sqrt{kT}\right)$ | $\tilde{O}\left(\|V\|^2\sqrt{T}\right)$ |
| tree graphs | $\tilde{O}\left(\|V\|^{\frac{5}{2}}\sqrt{T}\right)$ | $\tilde{O}\left(\|V\|^2\sqrt{T}\right)$ |
| grid graphs | $\tilde{O}\left(\|V\|^{\frac{5}{2}}\sqrt{T}\right)$ | $\tilde{O}\left(\|V\|^2\sqrt{T}\right)$ |
| complete graphs | $\tilde{O}\left(\|V\|^4\sqrt{T}\right)$ | $\tilde{O}\left(\|V\|^3\sqrt{T}\right)$ |

Table 1: Regret bound comparison with [25].

For each leaf $z$ under $y$, the event that $\mathrm{Path}(z)$ is the smallest live path from $s$ to $v$ is exclusive from each other. And that event is included in that $\mathrm{Path}(y)$ is the smallest live path from $s$ to $\mathrm{Node}(y)$. So

$$\sum_{z \text{ is a leaf under } y} p_{1,z}^{\boldsymbol{\mu}',S} p_{2,z}^{\boldsymbol{\mu}',S} \le p_{1,y}^{\boldsymbol{\mu}',S} p_{2,y}^{\boldsymbol{\mu}',S},$$

and thus

$$\sum_{z \text{ is a leaf under } y} \frac{p_{1,z}^{\boldsymbol{\mu}',S}}{p_{1,y}^{\boldsymbol{\mu}',S}} p_{2,z}^{\boldsymbol{\mu}',S} \le p_{2,y}^{\boldsymbol{\mu}',S} \le p_{2,y}^{\boldsymbol{\mu},S}.$$

So

$$(36) \le \left(\mu'_{\mathrm{Edge}(y)} - \mu_{\mathrm{Edge}(y)}\right) p_{1,\mathrm{Parent}(y)}^{\boldsymbol{\mu},S} p_{2,y}^{\boldsymbol{\mu},S}.$$

Then

$$(35) \le \sum_{y \text{ is a non-root node}} \left(\mu'_{\mathrm{Edge}(y)} - \mu_{\mathrm{Edge}(y)}\right) p_{1,\mathrm{Parent}(y)}^{\boldsymbol{\mu},S} p_{2,y}^{\boldsymbol{\mu},S} = \sum_{e \in E}(\mu'_e - \mu_e) \sum_{\mathrm{Edge}(y)=e} p_{1,\mathrm{Parent}(y)}^{\boldsymbol{\mu},S} p_{2,y}^{\boldsymbol{\mu},S}.$$

(37)

We then show

$$\sum_{\mathrm{Edge}(y)=e} p_{1,\mathrm{Parent}(y)}^{\boldsymbol{\mu},S} p_{2,y}^{\boldsymbol{\mu},S} \le p_e^{\boldsymbol{\mu},S}, \tag{38}$$

for every edge $e$. If $e$ is a directed edge from $u$ to $w$, $p_e^{\boldsymbol{\mu},S} \ge \sum_{\mathrm{Edge}(y)=e} p_{\mathrm{Parent}(y)}^{\boldsymbol{\mu},S}$, since $p_{\mathrm{Parent}(y)}^{\boldsymbol{\mu},S}$ is the probability that the path $\mathrm{Path}(\mathrm{Parent}(y))$ is the smallest live path from $s$ to $\mathrm{Node}(\mathrm{Parent}(y)) = u$, and thus such events are mutually exclusive for different $y$ with $\mathrm{Edge}(y) = e$. Then $p_e^{\boldsymbol{\mu},S} \ge \sum_{\mathrm{Edge}(y)=e} p_{1,\mathrm{Parent}(y)}^{\boldsymbol{\mu},S} p_{2,y}^{\boldsymbol{\mu},S}$ as $p_{2,\mathrm{Parent}(y)}^{\boldsymbol{\mu},S} \ge p_{2,y}^{\boldsymbol{\mu},S}$. Thus we have (38).

Combining Inequalities (37) and (38), we prove the key Inequality (32), which in turn shows that the influence maximization bandit satisfies the TPM bounded smoothness condition with $B = \max_{u \in V} |\{v \in V \mid v \text{ can be reached from } u\}|$.

## D    Detailed Comparison with [25] on the Regret Bounds for Influence Maximization Bandits

Let $G = (V, E)$ be the social graph we consider. By Lemma 2, our Theorem 1 can be applied to the influence maximization bandit with $B = \tilde{C} \le |V|$, which gives concrete $O(\log T)$ distribution-dependent and $O(\sqrt{T \log T})$ distribution-independent bounds for the influence maximization bandit. Wen et al. [25] also study the influence maximization bandit and eliminate the exponential factor $1/p^*$. They use a complexity term $C_*$ to characterize their regret bound, where $C_*$ has complicated relationship with network topology and edge probabilities. Wen et al. [25] list several families of graphs with concrete regret bounds, ignoring the effect of edge probabilities on their complexity term $C_*$. Our regret bounds with complexity term $\tilde{C}$ can also be applied to these graph families, and Table 1 list the comparison results between our regret bounds and their regret bounds. The

Figure 2: Reduction Structure

comparison shows that our regret bounds are always better than their bounds, with an improvement factor from $O(\sqrt{k})$ to $O(|V|)$, where $V$ is the set of nodes in the graph, and $k$ is the number of seeds to be selected in each round. This indicates that, in terms of characterizing the topology effect on the regret bound, our simple complexity term $\tilde{C}$ is more effective than their complicated term $C_*$.

# E  Lower Bound Proofs (for Section 5)

## E.1  Proof of Theorem 2

---
**Algorithm 2** Reduce MAB to CMAB-T
---
**Input:** $m, T_{\text{CMAB}}, p$ {$m$ is the number of arms, $T_{\text{CMAB}}$ is the number of rounds in CMAB, and $p$ is triggering probability.}
 1: **for** $t = 1, \ldots, T_{\text{CMAB}}$ **do**
 2:     sample $\gamma_t$ i.i.d. from Bernoulli distribution $B_p$
 3: **end for**
 4: $\mathcal{H} \leftarrow \varnothing; t_{\text{MAB}} \leftarrow 0$
 5: **for** $t = 1, \ldots, T_{\text{CMAB}}$ **do**
 6:     $S_{i_t} \leftarrow \mathsf{CMAB\text{-}Oracle}(\mathcal{H})$ {Oracle decides the CMAB-T action based on the execution history}
 7:     **if** $\gamma_t = 1$ **then**
 8:         $t_{\text{MAB}} \leftarrow t_{\text{MAB}} + 1$
 9:         In MAB, play arm $i_t$ in round $t_{\text{MAB}}$, obtain feedback $\tilde{X}^{(t_{\text{MAB}})}_{i_t}$
10:         In CMAB-T, $i_t$ is triggered with feedback $X^{(t)}_{i_t} = \tilde{X}^{(t_{\text{MAB}})}_{i_t}$, and set reward as $p^{-1} X^{(t)}_{i_t}$
11:         $\mathcal{H} \leftarrow \mathsf{Append}(\mathcal{H}, (S_{i_t}, \{i_t\}, X^{(t)}_{i_t}))$ {$\{i_t\}$ is the set of triggered arms}
12:     **else**
13:         {$\gamma_t = 0$, and MAB is not played in this case}
14:         In CMAB-T, no arm is triggered, and the reward is 0
15:         $\mathcal{H} \leftarrow \mathsf{Append}(\mathcal{H}, (S_{i_t}, \varnothing, -))$ {triggering set is empty, so no feedback}
16:     **end if**
17: **end for**{In the end, $T_{\text{MAB}} = t_{\text{MAB}}$}
---

We prove the theorem by reducing classical MAB to this CMAB-T game instance by Algorithm 2. For convenience, we define Bernoulli random variable $\gamma_t = \mathbb{I}\{\tau_t(S_{i_t}, X^{(t)}) = \{i_t\}\}$, where $S_{i_t}$ is the action played in round $t$, and thus $\gamma_t$ is an indicator representing whether a base arm is triggered

in round $t$. Moreover, to distinguish the environment outcome in MAB and CMAB-T in the reduction, we use $\tilde{X}^{(t_{\text{MAB}})}$ to denote the environment outcome in round $t_{\text{MAB}}$ of MAB, and $X^{(t)}$ to denote the environment outcome in round $t$ of CMAB-T.

Figure 2 shows the structure of reduction. Algorithm 2 adapts the CMAB-T algorithm to an MAB algorithm. Conversely, it also adapts the MAB instance to the corresponding CMAB-T instance. Thus when Algorithm 2 runs, we have one MAB instance and one CMAB-T instance running simultaneously. Let $T_{\text{CMAB}}$ be the total number of rounds in the CMAB-T instance and $T_{\text{MAB}}$ be the total number of rounds in the MAB instance. For convenience, we use $t$ to refer to the index of rounds in CMAB-T, while $t_{\text{MAB}}$ is the index of rounds in MAB. In Algorithm 2, we fix $T_{\text{CMAB}}$ and thus $T_{\text{MAB}}$ is a random variable. We have $T_{\text{MAB}} = \sum_{t=1}^{T_{\text{CMAB}}} \gamma_t$. So $\mathbb{E}[T_{\text{MAB}}] = pT_{\text{CMAB}}$ and we have following lemma about the distribution of $T_{\text{MAB}}$.

**Lemma 7.** *If $pT_{\text{CMAB}} \geq 6$, then $\Pr\left[T_{\text{MAB}} \geq \frac{1}{2}pT_{\text{CMAB}}\right] \geq \frac{1}{2}$.*

*Proof.* $T_{\text{MAB}} = \sum_{t=1}^{T_{\text{CMAB}}} \gamma_t$. By multiplicative Chernoff bound (Fact 2),

$$\Pr[T_{\text{MAB}} \geq \frac{1}{2}pT_{\text{CMAB}}] \geq 1 - \left(\frac{e^{-\frac{1}{2}}}{\left(\frac{1}{2}\right)^{\frac{1}{2}}}\right)^{pT_{\text{CMAB}}} \geq \frac{1}{2},$$

when $pT_{\text{CMAB}} \geq 6$.

$$\Pr[T_{\text{MAB}} \geq \frac{1}{2}pT_{\text{CMAB}}] \geq 1 - \left(e^{-\frac{1}{8}pT_{\text{CMAB}}}\right) \geq \frac{1}{2},$$

when $pT_{\text{CMAB}} \geq 6$.

In the following, we overload the notation $\mathcal{D}$ to also represent a probabilistic distribution of the environment instance (a.k.a. outcome distribution) $D$, and use $D \sim \mathcal{D}$ to represent a random environment instance $D$ drawn from the distribution $\mathcal{D}$.

**Lemma 8.** *Consider a random MAB environment instance $D$ drawn from a distribution $\mathcal{D}$. Assume we have a lower bound $L(T_{\text{MAB}})$ of expected regret, i.e. for every natural number $T_{\text{MAB}}$, any MAB algorithm $A$ has expected regret*

$$\mathbb{E}_{D\sim\mathcal{D}}[Reg^A_{\text{MAB},D}(T_{\text{MAB}})] \geq L(T_{\text{MAB}}).$$

*Then consider the corresponding CMAB-T environment instance $D$. For every natural number $T_{\text{CMAB}} \geq 5p^{-1}$, any CMAB-T algorithm $A$ has expected regret*

$$\mathbb{E}_{D\sim\mathcal{D}}[Reg^A_{\text{CMAB},D}(T_{\text{CMAB}})] \geq \frac{1}{2}p^{-1}L(\frac{1}{2}pT_{\text{CMAB}}). \tag{39}$$

*Proof.* Without loss of generality, we may assume $L(T)$ is non-decreasing, as regret of any strategy increases as $T$ increases.

We prove the lemma using the reduction described above. We run Algorithm 2 with $A$ be the CMAB-T oracle and $D$ be the environment instance. Let $\gamma$ be the vector $(\gamma_1, \gamma_2, \ldots, \gamma_{T_{\text{CMAB}}})$. Every possible value of $\gamma$ parameterizes Algorithm 2 into an algorithm plays MAB problem for $T_{\text{MAB}} = \sum_{t=1}^{T_{\text{CMAB}}} \gamma_t$ rounds. We denote this MAB algorithm with $A_\gamma$. By our assumption, $\mathbb{E}_{D\sim\mathcal{D}}[Reg^{A_\gamma}_{\text{MAB},D}(T_{\text{MAB}})] \geq L(T_{\text{MAB}})$.

Then we compare the regret in both cases. For a given distribution $D$, let $\mu_{i,D} = \mathbb{E}_{X\sim D}[X_i]$ and $\mu^*_D = \max_i \mu_{i,D}$. For MAB problem and every $\gamma$,

$$\mathbb{E}_{D\sim\mathcal{D}}[Reg^{A_\gamma}_{\text{MAB},D}(T_{\text{MAB}})] = \mathbb{E}_{D\sim\mathcal{D}}\left[T_{\text{MAB}} \cdot \mu^*_D - \mathbb{E}\left[\sum_{t=1}^{T_{\text{CMAB}}} \gamma_t X_{i_t}\right]\right]$$

$$= \mathbb{E}_{D\sim\mathcal{D}}\left[\mathbb{E}\left[\sum_{t=1}^{T_{\text{CMAB}}} \gamma_t(\mu^*_D - X_{i_t})\right]\right]$$

$$= \mathbb{E}_{D\sim\mathcal{D}}\left[\mathbb{E}\left[\sum_{t=1}^{T_{\text{CMAB}}} \gamma_t(\mu^*_D - \mu_{i_t,D})\right]\right],$$

where the inner expectation is taken over the rest randomness, including the randomness of $i_t$, which is based on the random feedback history and the possible randomness of algorithm $A_\gamma$. For CMAB-T, we have

$$\mathop{\mathbb{E}}_{D \sim \mathcal{D}}[Reg^A_{\text{CMAB},D}(T_{\text{CMAB}})]$$

$$= \mathop{\mathbb{E}}_{D \sim \mathcal{D}}\left[T_{\text{CMAB}} \cdot \mu^*_D - \mathop{\mathbb{E}}_{\gamma \sim B_p^{T_{\text{CMAB}}}}\left[\mathbb{E}\left[\sum_{t=1}^{T_{\text{CMAB}}} \gamma_t p^{-1} X_{i_t}\right]\right]\right]$$

$$= \mathop{\mathbb{E}}_{D \sim \mathcal{D}}\left[T_{\text{CMAB}} \cdot \mu^*_D - \mathop{\mathbb{E}}_{\gamma \sim B_p^{T_{\text{CMAB}}}}\left[\mathbb{E}\left[\sum_{t=1}^{T_{\text{CMAB}}} \gamma_t p^{-1} \mu_{i_t,D}\right]\right]\right]$$

$$= \mathop{\mathbb{E}}_{D \sim \mathcal{D}}\left[p T_{\text{CMAB}} \cdot p^{-1} \mu^*_D - \mathop{\mathbb{E}}_{\gamma \sim B_p^{T_{\text{CMAB}}}}\left[\mathbb{E}\left[\sum_{t=1}^{T_{\text{CMAB}}} \gamma_t p^{-1} \mu_{i_t,D}\right]\right]\right]$$

$$= \mathop{\mathbb{E}}_{D \sim \mathcal{D}}\left[\mathop{\mathbb{E}}_{\gamma \sim B_p^{T_{\text{CMAB}}}}\left[\sum_{t=1}^{T_{\text{CMAB}}} \gamma_t p^{-1} \mu^*_D\right] - \mathop{\mathbb{E}}_{\gamma \sim B_p^{T_{\text{CMAB}}}}\left[\mathbb{E}\left[\sum_{t=1}^{T_{\text{CMAB}}} \gamma_t p^{-1} \mu_{i_t,D}\right]\right]\right]$$

$$= p^{-1} \mathop{\mathbb{E}}_{D \sim \mathcal{D}, \gamma \sim B_p^{T_{\text{CMAB}}}}\left[\mathbb{E}\left[\sum_{t=1}^{T_{\text{CMAB}}} \gamma_t (\mu^* - \mu_{i_t,D})\right]\right],$$

where the innermost expectation is taken over the rest randomness such as the randomness of $i_t$. Therefore

$$\mathop{\mathbb{E}}_{D \sim \mathcal{D}}[Reg^A_{\text{CMAB},D}(T_{\text{CMAB}})] = p^{-1} \mathop{\mathbb{E}}_{D \sim \mathcal{D}, \gamma \sim B_p^{T_{\text{CMAB}}}}[Reg^{A_\gamma}_{\text{MAB},D}(T_{\text{MAB}})].$$

Calculation above also shows $\mathbb{E}_{D \sim \mathcal{D}}[Reg^{A_\gamma}_{\text{MAB},D}(T_{\text{MAB}})] \geq 0$. And by monotonicity of $L(T)$,

$$\mathop{\mathbb{E}}_{D}[Reg^A_{\text{CMAB},D}(T_{\text{CMAB}})] = p^{-1} \mathop{\mathbb{E}}_{D,\gamma}[Reg^{A_\gamma}_{\text{MAB},D}(T_{\text{MAB}})]$$

$$\geq p^{-1} \mathop{\mathbb{E}}_{D,\gamma}[\mathbb{I}\{T_{\text{MAB}} \geq \frac{1}{2} p T_{\text{CMAB}}\} Reg^{A_\gamma}_{\text{MAB},D}(T_{\text{MAB}})]$$

$$\geq p^{-1} \mathop{\mathbb{E}}_{D,\gamma}[\mathbb{I}\{T_{\text{MAB}} \geq \frac{1}{2} p T_{\text{CMAB}}\} L(\frac{1}{2} p T_{\text{CMAB}})]$$

$$= p^{-1} \mathop{\Pr}_{D,\gamma}\{T_{\text{MAB}} \geq \frac{1}{2} p T_{\text{CMAB}}\} L(\frac{1}{2} p T_{\text{CMAB}})$$

$$\geq \frac{1}{2} p^{-1} L(\frac{1}{2} p T_{\text{CMAB}}). \qquad \square$$

**Lemma 9.** *Let $m$ be the number of arms and $T$ be the number of rounds. Let $\varepsilon = \frac{1}{10}\sqrt{m/T}$. Then define the family of MAB outcome distributions $\mathcal{D} = \{D_1, \dots, D_m\}$ with*

$$\mathop{\Pr}_{D_j}\{X_i = 1\} = \begin{cases} \frac{1}{2} & \text{if } i \neq j \\ \frac{1}{2} + \varepsilon & \text{if } i = j \end{cases}.$$

*Let $D$ be a random environment instance uniformly drawn from $\mathcal{D}$, then for any MAB algorithm $A$,*

$$\mathop{\mathbb{E}}_{D \sim \mathcal{D}}[Reg^A_{\text{MAB},D}(T)] \geq \frac{\varepsilon T}{6} = \frac{1}{60}\sqrt{mT}.$$

*Proof of Theorem 2.* Let $\mathcal{D}$ be the family of outcome distributions defined in Lemma 9, and $D$ is uniformly drawn from $\mathcal{D}$. Applying the result of Lemma 9 to Lemma 8, with $L(T) = \frac{1}{60}\sqrt{mT}$ in Lemma 8, we have

$$\mathop{\mathbb{E}}_{D \sim \mathcal{D}}[Reg^A_{\text{CMAB},D}(T)] \geq \frac{1}{2} p^{-1} L(\frac{1}{2} p T)$$

$$= \frac{1}{2} p^{-1} \cdot \frac{1}{60}\sqrt{\frac{1}{2} m p T}$$

$$> \frac{1}{170}\sqrt{\frac{mT}{p}}.$$

Since $D$ is uniformly drawn from $\mathcal{D}$, then there must exists a $D \in \mathcal{D}$ such that

$$Reg^A_{\text{CMAB},D}(T) \geq \frac{1}{170}\sqrt{\frac{mT}{p}}. \qquad \square$$

It is easy to show corresponding CMAB-T problem satisfies original bounded smoothness (Condition 5) with $f(x) = x$. So the theorem above gives an example that the upper bound in [7] is tight up to a $O(\sqrt{\log T})$ factor.

### E.2   Proof of Theorem 3

*Proof of Theorem 3.* We regard this kind of CMAB-T problem instances as a variant of classical MAB, that each arm gives three possible outcomes, 0, 1, and $\perp$. Denote these arms with random variables $X'_1, \ldots, X'_n$. The reward is $p^{-1}$ times of the outcome if the outcome is 0 or 1, while the reward is 0 if the outcome is $\perp$. This variant is equivalent to the CMAB-T instances: Outcome $X'_i = \perp$ corresponds to Bernoulli base arm $X_i$ in CMAB-T not being triggered, outcome $X'_i = 1$ or 0 corresponds to Bernoulli base arm $X_i$ being triggered and $X_i = 1$ or 0, respectively. Thus $\Pr[X'_i = \perp] = 1 - p$, $\Pr[X'_i = 0] = p(1 - \mu_i)$, and $\Pr[X'_i = 1] = p\mu_i$, where $p$ is the triggering probability and $\mu_i$ is the expectation of $X_i$.

Let $X$ and $Y$ be random variables whose values are in the same finite set $V$. Define the KL-divergence

$$\text{kl}(X, Y) = \sum_{x \in V} \Pr\{X = x\} \ln \frac{\Pr\{X = x\}}{\Pr\{Y = x\}}.$$

For example the KL-divergence between $X'_1$ and $X'_2$ is

$$\begin{aligned}
\text{kl}(X'_1, X'_2) &= \Pr\{X'_1 = \perp\} \ln \frac{\Pr\{X'_1 = \perp\}}{\Pr\{X'_2 = \perp\}} + \Pr\{X'_1 = 0\} \ln \frac{\Pr\{X'_1 = 0\}}{\Pr\{X'_2 = 0\}} \\
&\quad + \Pr\{X'_1 = 1\} \ln \frac{\Pr\{X'_1 = 1\}}{\Pr\{X'_2 = 1\}} \\
&= (1 - p) \ln \frac{1 - p}{1 - p} + p(1 - \mu_1) \ln \frac{p(1 - \mu_1)}{p(1 - \mu_2)} + p\mu_1 \ln \frac{p\mu_1}{p\mu_2} \\
&= 0 + p(1 - \mu_1) \ln \frac{1 - \mu_1}{1 - \mu_2} + p\mu_1 \ln \frac{\mu_1}{\mu_2} \\
&= p \cdot \left[ (1 - \mu_1) \ln \frac{1 - \mu_1}{1 - \mu_2} + \mu_1 \ln \frac{\mu_1}{\mu_2} \right] \\
&= p \cdot \text{kl}(X_1, X_2). \qquad \square
\end{aligned}$$

Thus, intuitively it takes $p^{-1}$ times more rounds to differentiate $X'_1$ and $X'_2$ than $X_1$ and $X_2$, which is stated formally in theorem below.

*Proof.* The analysis is generalized from the case that the arms are Bernoulli random variables. For an arm $i$, we use $N_i(T)$ to denote the number of times the arm $i$ is played in $T$ rounds. For each non-optimal arm $i$, i.e. $\mu_i < \mu^* < 1$, we show

$$\liminf_{T \to +\infty} \frac{\mathbb{E}[N_i(T)]}{\ln T} \geq \frac{p^{-1}}{\text{kl}(X_i, X_{i^*})} = \frac{1}{\text{kl}(X'_i, X'_{i^*})}. \qquad (40)$$

Then by formula

$$Reg^A_{\boldsymbol{\mu}}(T) = \sum_{i : \mu_i < \mu^*} \mathbb{E}[N_i(T)]\Delta_i,$$

the theorem holds.

Without loss of generality, we may assume arm 1 is an optimal arm and arm 2 is non-optimal. We prove Eq. (40) for arm 2 and then the inequality holds for every arm. Consider that if we replace arm 2 with a fictional arm $2'$, which has an expectation $\mu_{2'}$ slightly greater than $\mu_1$, then arm 1 will

become non-optimal and strategy $A$ will play arm 1 for $o(n^a)$ times for any $a > 0$. So strategy $A$ must play arm 2 for enough times, to differentiate from arm $2'$.

Formally, let $\varepsilon > 0$ be any positive real number. Let $\mu_{2'}$ be a real number such that $\mu_{2'} > \mu_1$ and

$$\text{kl}(X_2, X_{2'}) = (1 - \mu_2)\ln\frac{1 - \mu_2}{1 - \mu_{2'}} + \mu_2\ln\frac{\mu_2}{\mu_{2'}} < (1 + \varepsilon)\text{kl}(X_2, X_1). \tag{41}$$

There exists such $\mu_{2'}$, because the left hand side of (41) is continuous as a function of $\mu_{2'}$. We use $\mathbb{E}'$ and $\Pr'$ to denote expectation and probability in the circumstance that arm $X_2$ is replaced by arm $X_{2'}$.

We define the empirical KL-divergence after the first $s$ samples of the arm $2/2'$,

$$\widehat{\text{kl}}_s = \sum_{t=1}^{s} Y_t,$$

where

$$Y_t = \begin{cases} \ln\frac{1 - \mu_2}{1 - \mu_{2'}}, & \text{if } X'_{2,t} = 0, \\ \ln\frac{\mu_2}{\mu_{2'}}, & \text{if } X'_{2,t} = 1, \\ 0, & \text{if } X'_{2,t} = \perp. \end{cases}$$

and $X'_{2,t}$ is result of the $t$-th sample of arm $2/2'$. Note that $(Y_t)$ are independent and $\mathbb{E}[Y_t] = \text{kl}(X'_2, X'_{2'})$.

First we prove

$$\Pr\left\{ N_2(T) < \frac{1 - \varepsilon}{\text{kl}(X'_2, X'_{2'})}\ln T \wedge \widehat{\text{kl}}_{N_2(T)} \le \left(1 - \frac{\varepsilon}{2}\right)\ln T \right\} = o(1). \tag{42}$$

We use the shorthands

$$C_T = \left\{ N_2(T) < \frac{1 - \varepsilon}{\text{kl}(X'_2, X'_{2'})}\ln T \wedge \widehat{\text{kl}}_{N_2(T)} \le \left(1 - \frac{\varepsilon}{2}\right)\ln T \right\}, \tag{43}$$

and

$$f_T = \frac{1 - \varepsilon}{\text{kl}(X'_2, X'_{2'})}\ln T.$$

If arm 2 is replaced by arm $2'$, we have

$$\Pr'\{C_T\} \le \Pr'\{N_2(T) < f_T\} \le \frac{\mathbb{E}'[T - N_2(T)]}{T - f_T},$$

where the second inequality is due to Markov's inequality. Recall the definition of consistent strategy, as $2'$ is the only optimal arm, we have $\mathbb{E}'[T - N_2(T)] = o(T^{\frac{\varepsilon}{2}})$. And by $T - f_T = \Omega(T)$, $\Pr'\{C_T\} = o(T^{\frac{\varepsilon}{2}-1})$. Then we use the property of KL-divergence

$$\Pr\{C_T\} = \mathbb{E}'\left[\mathbb{I}\{C_T\} \cdot \exp\left(\widehat{\text{kl}}_{N_2(T)}\right)\right],$$

then

$$\Pr\{C_T\} = \mathbb{E}'\left[\mathbb{I}\{C_n\} \cdot \exp\left(\widehat{\text{kl}}_{N_2(T)}\right)\right] \le \Pr'\{C_T\}\cdot\exp\left[\left(1 - \frac{\varepsilon}{2}\right)\ln T\right] = \Pr'\{C_T\}\cdot T^{1-\frac{\varepsilon}{2}} = o(1).$$

Second, we prove

$$\Pr\left\{ N_2(T) < f_T \wedge \widehat{\text{kl}}_{T_2(T)} > \left(1 - \frac{\varepsilon}{2}\right)\ln T \right\} = o(1). \tag{44}$$

We have

$$\Pr\left\{ N_2(T) < f_T \wedge \widehat{\text{kl}}_{N_2(T)} > \left(1 - \frac{\varepsilon}{2}\right)\ln T \right\} \le \Pr\left\{ N_2(T) < f_T \wedge \max_{s \le f_T}\widehat{\text{kl}}_s > \left(1 - \frac{\varepsilon}{2}\right)\ln T \right\}$$

$$\le \Pr\left\{ \max_{s \le f_T}\widehat{\text{kl}}_s > \left(1 - \frac{\varepsilon}{2}\right)\ln T \right\}.$$

Recall the definition of $\widehat{\mathrm{kl}}_s$, which is a summation of independent random variables with the same distribution over a finite support, whose expectation is $\mathrm{kl}(X_2', X_{2'}')$. So we apply the maximal version of the strong law of large numbers, and then (44) holds, as $f_T \cdot \mathrm{kl}(X_2', X_{2'}') = (1 - \varepsilon) \ln T$.

In conclusion, combining Eq. (42) and (44), we have $\Pr\{N_2(T) < f_T\} = o(1)$, implying

$$
\begin{aligned}
\mathbb{E}\left[N_2(T)\right] &\geq (1 - o(1)) \cdot f_T \\
&= (1 - o(1)) \cdot \frac{1 - \varepsilon}{\mathrm{kl}(X_2', X_{2'}')} \ln T \\
&\geq (1 - o(1)) \cdot \frac{1 - \varepsilon}{1 + \varepsilon} \frac{\ln T}{\mathrm{kl}(X_2', X_1')}.
\end{aligned}
$$

Then (40) holds, as $\varepsilon$ can be any positive real number, and thus the theorem holds. $\qquad\square$

# F    Results with $\infty$-norm TPM Conditions

## F.1    TPM Conditions with the $\infty$-norm

We first restate the original bounded smoothness condition in [7] below, which is an $\infty$-norm based condition.

**Condition 5 (Bounded Smoothness).** *We say that a CMAB-T problem instance satisfies* bounded smoothness*, if there exists a continuous, strictly increasing (and thus invertible) function $f(\cdot)$ with $f(0) = 0$, such that for any two distributions $D, D' \in \mathcal{D}$ with expectation vectors $\boldsymbol{\mu} = (\mu_1, \ldots, \mu_m)$ and $\boldsymbol{\mu}' = (\mu_1', \ldots, \mu_m')$, and for any $\Lambda > 0$, we have $|r_{\boldsymbol{\mu}}(S) - r_{\boldsymbol{\mu}'}(S)| \leq f(\Lambda)$ if $\max_{i \in \tilde{S}} |\mu_i - \mu_i'| \leq \Lambda$, for all $S \in \mathcal{S}$, where $\tilde{S} = \{i \in [m] \mid \Pr_{X \sim D, \tau}\{i \in \tau(S, X)\} > 0\}$ is the set of arms that could be triggered by action $S$.*

Note that $f(\cdot)$ may depend on problem instance parameters such as $m$, but not on action $S$ or mean vectors $\boldsymbol{\mu}, \boldsymbol{\mu}'$.

Similar to the 1-norm case, we use triggering probabilities to modulate the bounded smoothness condition to obtain the following TPM version:

**Condition 6. ($\infty$-Norm TPM Bounded Smoothness)** *We say a CMAB-T problem instance satisfies the* triggering-probability-modulated (TPM) bounded smoothness *with bounded smoothness function $f(x)$, if for any two distributions $D, D' \in \mathcal{D}$ with expectation vectors $\boldsymbol{\mu}$ and $\boldsymbol{\mu}'$, any action $S$ and any $\Lambda > 0$, we have $|r_S(\boldsymbol{\mu}) - r_S(\boldsymbol{\mu}')| \leq f(\Lambda)$ if $\max_{i \in [m]} p_i^{D,S} |\mu_i - \mu_i'| \leq \Lambda$.*

Note that Condition 6 is stronger than Condition 5 under the same bounded smoothness function $f$. This is because if we have $\max_{i \in [m]} |\mu_i - \mu_i'| \leq \Lambda$, then we have $\max_{i \in [m]} p_i^{D,S} |\mu_i - \mu_i'| \leq \Lambda$. Then if Condition 6 holds, we have $|r_S(\boldsymbol{\mu}) - r_S(\boldsymbol{\mu}')| \leq f(\Lambda)$. This means that if Condition 6 holds, we have $|r_S(\boldsymbol{\mu}) - r_S(\boldsymbol{\mu}')| \leq f(\Lambda)$ if $\max_{i \in [m]} |\mu_i - \mu_i'| \leq \Lambda$, which is exactly Condition 5.

## F.2    Theorem and Proofs with $\infty$-norm TPM Conditions

**Theorem 5.** *Suppose a CMAB-T problem instance $([m], \mathcal{S}, \mathcal{D}, D^{\mathrm{trig}}, R)$ satisfies monotonicity (Condition 1). For a fixed environment instance $D \in \mathcal{D}$ with expectation vector $\boldsymbol{\mu}$, the $T$-round $(\alpha, \beta)$-approximation regret bound using an $(\alpha, \beta)$-approximation oracle in various cases are given below.*

    *(1) For the CUCB algorithm on a problem instance that satisfies TPM bounded smoothness (Condition 6) with bounded smoothness function $f(x)$, together with $\Delta_{\min} > 0$, the regret is at most*

$$
\sum_{i \in [m]} 78 \ln T \left( \frac{\Delta_{\min}^i}{f^{-1}(\Delta_{\min}^i)^2} + \int_{\Delta_{\min}^i}^{\Delta_{\max}^i} \frac{1}{f^{-1}(x)^2} \, \mathrm{d}x \right)
$$

$$
+ m \cdot \left[ \left( \frac{\pi^2}{6} + 1 \right) \lceil -\log_2 f^{-1}(\Delta_{\min}) \rceil_0 + \frac{\pi^2}{3} + 1 \right] \cdot \Delta_{\max};
$$

*(2) For the CUCB algorithm on a problem instance that satisfies TPM bounded smoothness (Condition 6) with bounded smoothness function $f(x) = ax$, the regret is at most*

$$25a\sqrt{mT\ln T} + m \cdot \left[\left(\frac{\pi^2}{6} + 1\right)\left\lceil -\log_2(\sqrt{156m\ln T/T})\right\rceil_0 + \frac{\pi^2}{3} + 1\right] \cdot \Delta_{\max};$$

We have several remarks on Theorem 5. First, the condition $\Delta_{\min} > 0$ automatically holds if the action space $\mathcal{S}$ is finite. Thus it is not an extra condition comparing to the result in [7] when actions are set of base arms. If $\Delta_{\min}$ is zero due to infinite $\mathcal{S}$, then we do not have regret bound as in (1), but we still have regret bound as in (2). Second, the regret bound in (1) is distribution-dependent bound, since it depends on $\Delta_{\min}^i$, which is determined by the distribution $D$; regret bounds in (2) is distribution-independent bound, since $\Delta_{\max}$ can be easily replaced by a quantity only depending on the problem instance, such as the maximum possible reward value. Third, when $\Delta_{\min}^i = +\infty$, $\frac{\Delta_{\min}^i}{f^{-1}(\Delta_{\min}^i)^2} = 0$.

### F.2.1 Proof of Theorem 5

In this subsection, we focus on giving a roadmap to prove Theorem 5 and showing the new techniques we invented to improve the regret bound. The remaining part of the proof is roughly the new calculation based on the old techniques (c.f. [7]).

In this subsection, we omit $(\alpha, \beta)$-approximation for clarity, in other words, we assume $\alpha = \beta = 1$. Generalization to accommodate $(\alpha, \beta)$ approximation can be found in the discussion section.

To exploit the advantage of TPM bounded smoothness condition (Conditions 6), for each arm $i$, we divide actions into groups according to $p_i^{D,S}$.

For convenience, we also allow to index the counters with $q_i^{D,S_t} > 0$, such that $N_{i,q_i^{D,S_t}}$ indicates the same counter as $N_{i,j}$ with $q_i^{D,S_t} = 2^{-j}$.

We use a shorthand as follows. For every arm $i$ and action $S$, define

$$q_i^{D,S} = \begin{cases} 2^{-j}, & \text{if } S \in \mathcal{S}_{i,j}^D, \\ 0, & \text{if } p_i^{D,S} = 0. \end{cases}$$

**Definition 8.**

$$\ell_t(\Delta, q) = \begin{cases} 0, & \text{if } q \leq \frac{1}{2}f^{-1}(\Delta), \\ \lfloor \frac{6\ln t}{f^{-1}(\Delta)^2} \rfloor + 1, & \text{if } q = 1, \\ \lfloor \frac{72q\ln t}{f^{-1}(\Delta)^2} \rfloor + 1, & \text{otherwise.} \end{cases}$$

To unify the proofs for distribution-dependent and distribution-independent bounds, we introduce a positive real number $M$. To prove the distribution-dependent bound, we will let $M = \Delta_{\min}$ or $M = \Delta_{\min}^i$ in some circumstances. To prove the distribution-independent bound, we will let $M = \tilde{\Theta}(T^{-1/2})$ to balance bounds for $Reg(\{\Delta_{S_t} \geq M\})$ and $Reg(\{\Delta_{S_t} < M\})$. And we implement $\mathcal{N}_t^i$ (Definition 7) with $j_{\max}^i = j_{\max}(M) = \lceil -\log_2 f^{-1}(M)\rceil_0$ The following are three technical claims used in the main proof, and we define the proofs of these claims to Section F.2.2.

**Claim 1 (Bound of insufficiently sampled regret).** *For any CMAB-T problem instance, any bounded smoothness function $f(x)$, any algorithm, any arm $i$, any natural number $j$ and any positive real number $M$,*

$$Reg(\{\Delta_{S_t} \geq M, S_t \in \mathcal{S}_{i,j}, N_{i,j,t-1} < \ell_T(\Delta_{S_t}, 2^{-j})\}) \leq \ell_T(M, 2^{-j})M + \int_M^{\max\{\Delta_{\max}^i, M\}} \ell_T(x, 2^{-j})\,\mathrm{d}x.$$

**Claim 2 (Bound of sufficiently sampled regret for CUCB).** *For the CUCB algorithm on a problem instance that satisfies TPM bounded smoothness (Condition 6) with bounded smoothness function $f(x)$,*

$$Reg(\{\Delta_{S_t} \geq M, \forall i, N_{i,q_i^{S_t},t-1} \geq \ell_T(\Delta_{S_t}, q_i^{S_t})\}) \leq m \cdot (\lceil -\log_2 f^{-1}(M)\rceil_0 + 2) \cdot \frac{\pi^2}{6} \cdot \Delta_{\max}.$$

We continue the proof of Theorem 5. Fix a value $M > 0$, we have

$$
\begin{aligned}
Reg(\{\}) &= Reg(\{\Delta_{S_t} < M\}) + Reg(\{\Delta_{S_t} \geq M\}) \\
&= Reg(\{\Delta_{S_t} < M\}) + Reg(\{\Delta_{S_t} \geq M, \forall i, N_{i,q_i^{S_t},t-1} \geq \ell_T(\Delta_{S_t}, q_i^{S_t})\}) \\
&\quad + Reg(\{\Delta_{S_t} \geq M, \exists i, N_{i,q_i^{S_t},t-1} < \ell_T(\Delta_{S_t}, q_i^{S_t})\}) \\
&\leq Reg(\{\Delta_{S_t} < M\}) + Reg(\{\Delta_{S_t} \geq M, \forall i, N_{i,q_i^{S_t},t-1} \geq \ell_T(\Delta_{S_t}, q_i^{S_t})\}) \\
&\quad + \sum_{i \in [m]} Reg(\{\Delta_{S_t} \geq M, N_{i,q_i^{S_t},t-1} < \ell_T(\Delta_{S_t}, q_i^{S_t})\}) \\
&\leq Reg(\{\Delta_{S_t} < M\}) + Reg(\{\Delta_{S_t} \geq M, \forall i, N_{i,q_i^{S_t},t-1} \geq \ell_T(\Delta_{S_t}, q_i^{S_t})\}) \\
&\quad + \sum_{i \in [m]} \sum_{j \geq 0} Reg(\{\Delta_{S_t} \geq M, S_t \in \mathcal{S}_{i,j}, N_{i,q_i^{S_t},t-1} < \ell_T(\Delta_{S_t}, q_i^{S_t})\}) \\
&= Reg(\{\Delta_{S_t} < M\}) + Reg(\{\Delta_{S_t} \geq M, \forall i, N_{i,q_i^{S_t},t-1} \geq \ell_T(\Delta_{S_t}, q_i^{S_t})\}) \\
&\quad + \sum_{i \in [m]} \sum_{j \geq 0} Reg(\{\Delta_{S_t} \geq M, S_t \in \mathcal{S}_{i,j}, N_{i,j,t-1} < \ell_T(\Delta_{S_t}, 2^{-j})\}). \quad (45)
\end{aligned}
$$

For the last part, if $j \geq \lceil -\log_2 f^{-1}(M) \rceil_0 + 1$, then $2^{-j} \leq \frac{1}{2} f^{-1}(M)$ and

$$
\frac{1}{2} f^{-1}(\Delta_{S_t}) \geq \frac{1}{2} f^{-1}(M) \geq 2^{-j}.
$$

By Definition 8, $\ell_T(\Delta_{S_t}, 2^{-j}) = 0$. Then $N_{i,j,t-1} < \ell_T(\Delta_{S_t}, 2^{-j})$ is impossible, so

$$
\sum_{j \geq \lceil -\log_2 f^{-1}(M) \rceil_0 + 1} Reg(\{\Delta_{S_t} \geq M, S_t \in \mathcal{S}_{i,j}, N_{i,j,t-1} < \ell_T(\Delta_{S_t}, 2^{-j})\}) = 0.
$$

**Lemma 10.** *For every arm $i$, the event-filtered regret*

$$
\sum_{j \geq 0} Reg(\{\Delta_{S_t} \geq M, S_t \in \mathcal{S}_{i,j}, N_{i,j,t-1} < \ell_T(\Delta_{S_t}, 2^{-j})\}) \quad (46)
$$

$$
\leq 78 \ln T \left( \frac{M}{f^{-1}(M)^2} + \int_M^{\max\{\Delta_{\max}^i, M\}} \frac{1}{f^{-1}(x)^2} \, \mathrm{d}x \right) + (j_{\max}(M) + 1) \cdot \Delta_{\max}^i.
$$

*Proof.* If $M > \Delta_{\max}^i$, it is impossible to have $\Delta_{S_t} \geq M$ and $S_t \in \mathcal{S}_{i,j}$ at the same time and then (46) $= 0$. Then the lemma holds trivially. So we may assume that $M \leq \Delta_{\max}^i$. By Claim 1,

$$
\begin{aligned}
(46) &= \sum_{j=0}^{j_{\max}(M)} Reg(\{\Delta_{S_t} \geq M, S_t \in \mathcal{S}_{i,j}, N_{i,j,t-1} < \ell_T(\Delta_{S_t}, 2^{-j})\}) \\
&\leq \sum_{j=0}^{j_{\max}(M)} \left( \ell_T(M, 2^{-j}) M + \int_M^{\max\{\Delta_{\max}^i, M\}} \ell_T(x, 2^{-j}) \, \mathrm{d}x \right) \\
&= \sum_{j=0}^{j_{\max}(M)} \left( \ell_T(M, 2^{-j}) M + \int_M^{\Delta_{\max}^i} \ell_T(x, 2^{-j}) \, \mathrm{d}x \right) \\
&= \sum_{j=0}^{j_{\max}(M)} \ell_T(M, 2^{-j}) M + \int_M^{\Delta_{\max}^i} \sum_{j=0}^{j_{\max}(M)} \ell_T(x, 2^{-j}) \, \mathrm{d}x. \quad (47)
\end{aligned}
$$

We then expand the notation $\ell_T(\Delta, q)$ (c.f. Definition 8) with

$$
\ell_T(\Delta, q) \leq \begin{cases} \frac{6 \ln T}{f^{-1}(\Delta)^2} + 1, & \text{if } q = 1, \\ \frac{72q \ln T}{f^{-1}(\Delta)^2} + 1, & \text{otherwise.} \end{cases}
$$

So for any $x \in [M, \Delta^i_{\max}]$,

$$\sum_{j=0}^{j_{\max}(M)} \ell_T(x, 2^{-j}) = \ell_T(x, 1) + \sum_{j=1}^{j_{\max}(M)} \ell_T(x, 2^{-j})$$

$$\leq \left( \frac{6 \ln T}{f^{-1}(x)^2} + 1 \right) + \sum_{j=1}^{j_{\max}(M)} \left( \frac{72 \cdot 2^{-j} \ln T}{f^{-1}(x)^2} + 1 \right)$$

$$= \frac{6 \ln T}{f^{-1}(x)^2} + \sum_{j=1}^{j_{\max}(M)} \frac{72 \cdot 2^{-j} \ln T}{f^{-1}(x)^2} + j_{\max}(M) + 1$$

$$\leq \frac{6 \ln T}{f^{-1}(x)^2} + \frac{72 \ln T}{f^{-1}(x)^2} + j_{\max}(M) + 1$$

$$= \frac{78 \ln T}{f^{-1}(x)^2} + j_{\max}(M) + 1.$$

Then we continue (47) with

$$(47) \leq \left( \frac{78 \ln T}{f^{-1}(M)^2} + j_{\max}(M) + 1 \right) \cdot M + \int_M^{\Delta^i_{\max}} \left( \frac{78 \ln T}{f^{-1}(x)^2} + j_{\max}(M) + 1 \right) \, \mathrm{d}x$$

$$= \frac{78 \ln T}{f^{-1}(M)^2} \cdot M + \int_M^{\Delta^i_{\max}} \frac{78 \ln T}{f^{-1}(x)^2} \, \mathrm{d}x + (j_{\max}(M) + 1) \cdot \Delta^i_{\max}$$

$$= 78 \ln T \left( \frac{M}{f^{-1}(M)^2} + \int_M^{\Delta^i_{\max}} \frac{1}{f^{-1}(x)^2} \, \mathrm{d}x \right) + (j_{\max}(M) + 1) \cdot \Delta^i_{\max}.$$

Hence the lemma holds. $\qquad \square$

**Lemma 11.** *For event-filtered regret*

$$Reg(\{\Delta_{S_t} < M\}) + \sum_{i \in [m]} \sum_{j \geq 0} Reg(\{\Delta_{S_t} \geq M, S_t \in \mathcal{S}_{i,j}, N_{i,j,t-1} < \ell_T(\Delta_{S_t}, 2^{-j})\}), \quad (48)$$

*(1) take $M = \Delta_{\min}$ when $\Delta_{\min} > 0$,*

$$(48) \leq \sum_{i \in [m]} 78 \ln T \left( \frac{\Delta^i_{\min}}{f^{-1}(\Delta^i_{\min})^2} + \int_{\Delta^i_{\min}}^{\Delta^i_{\max}} \frac{1}{f^{-1}(x)^2} \, \mathrm{d}x \right) + m \cdot (j_{\max}(\Delta_{\min}) + 1) \cdot \Delta_{\max};$$

*(2) if $f(x) = ax$, then take $M = a\sqrt{156m \ln T / T}$,*

$$(48) < 25a\sqrt{mT \ln T} + m \cdot (j_{\max}(a\sqrt{156m \ln T / T}) + 1) \cdot \Delta_{\max}.$$

*Proof.* (1) If $\Delta_{S_t} < M = \Delta_{\min}$, then $\Delta_{S_t} = 0$. So $Reg(\{\Delta_{S_t} < M\}) \leq 0$. For every $i \in [m]$ and every integer $j$, we may replace $M$ with $\Delta^i_{\min}$ as below.

$$Reg(\{\Delta_{S_t} \geq M, S_t \in \mathcal{S}_{i,j}, N_{i,j,t-1} < \ell_T(\Delta_{S_t}, 2^{-j})\}) \quad (49)$$

$$= Reg(\{\Delta_{S_t} \geq \Delta_{\min}, S_t \in \mathcal{S}_{i,j}, N_{i,j,t-1} < \ell_T(\Delta_{S_t}, 2^{-j})\})$$

$$= Reg(\{\Delta_{S_t} \geq \Delta^i_{\min}, S_t \in \mathcal{S}_{i,j}, N_{i,j,t-1} < \ell_T(\Delta_{S_t}, 2^{-j})\}).$$

Then apply Lemma 10 with $M = \Delta^i_{\min}$, we have

$$(48) = \sum_{i \in [m]} \sum_{j \geq 0} Reg(\{\Delta_{S_t} \geq M, S_t \in \mathcal{S}_{i,j}, N_{i,j,t-1} < \ell_T(\Delta_{S_t}, 2^{-j})\})$$

$$\leq \sum_{i \in [m]} \left[ 78 \ln T \left( \frac{\Delta^i_{\min}}{f^{-1}(\Delta^i_{\min})^2} + \int_{\Delta^i_{\min}}^{\Delta^i_{\max}} \frac{1}{f^{-1}(x)^2} \, \mathrm{d}x \right) + (j_{\max}(\Delta^i_{\min}) + 1) \cdot \Delta^i_{\max} \right]$$

$$\leq \sum_{i \in [m]} 78 \ln T \left( \frac{\Delta^i_{\min}}{f^{-1}(\Delta^i_{\min})^2} + \int_{\Delta^i_{\min}}^{\Delta^i_{\max}} \frac{1}{f^{-1}(x)^2} \, \mathrm{d}x \right) + m \cdot (j_{\max}(\Delta_{\min}) + 1) \cdot \Delta_{\max}.$$

(2) By Lemma 10, for every arm $i$,

$$\sum_{j\geq 0} Reg(\{\Delta_{S_t} \geq M, S_t \in \mathcal{S}_{i,j}, N_{i,j,t-1} < \ell_T(\Delta_{S_t}, 2^{-j})\})$$

$$\leq 78\ln T \left( \frac{M}{f^{-1}(M)^2} + \int_M^{\Delta_{\max}^i} \frac{1}{f^{-1}(x)^2}\, \mathrm{d}x \right) + (j_{\max}(M) + 1)\cdot \Delta_{\max}^i$$

$$= 78\ln T \left( \frac{M}{(a^{-1}M)^2} + \int_M^{\Delta_{\max}^i} \frac{1}{(a^{-1}x)^2}\, \mathrm{d}x \right) + (j_{\max}(M) + 1)\cdot \Delta_{\max}^i$$

$$= 78\ln T \left( \frac{1}{a^{-2}M} + \int_M^{\Delta_{\max}^i} \frac{1}{a^{-2}x^2}\, \mathrm{d}x \right) + (j_{\max}(M) + 1)\cdot \Delta_{\max}^i$$

$$\leq 78\ln T \left( \frac{1}{a^{-2}M} + \frac{1}{a^{-2}M} \right) + (j_{\max}(M) + 1)\cdot \Delta_{\max}^i$$

$$= \frac{156\ln T}{a^{-2}M} + (j_{\max}(M) + 1)\cdot \Delta_{\max}. \tag{50}$$

$Reg(\{\Delta_{S_t} < M\}) < TM$ as the regret in each round is less than $M$. So by (50) and take $M = a\sqrt{156m\ln T/T}$,

$$(48) < TM + \frac{156m\ln T}{a^{-2}M} + m\cdot (j_{\max}(M) + 1)\cdot \Delta_{\max}$$

$$= a\sqrt{156mT\ln T} + a\sqrt{156mT\ln T} + m\cdot (j_{\max}(M) + 1)\cdot \Delta_{\max}$$

$$< 25a\sqrt{mT\ln T} + m\cdot (j_{\max}(a\sqrt{156m\ln T/T}) + 1)\cdot \Delta_{\max}. \qquad \square$$

*Proof of Theorem 5.* (1) Since $\Delta_{\min} > 0$, we can take $M = \Delta_{\min}$. By Lemma 11(1) and Claim 2, we continue Inequality (45) as below.

$$(45) \leq \sum_{i\in[m]} 78\ln T \left( \frac{\Delta_{\min}^i}{f^{-1}(\Delta_{\min}^i)^2} + \int_{\Delta_{\min}^i}^{\Delta_{\max}^i} \frac{1}{f^{-1}(x)^2}\, \mathrm{d}x \right) + m\cdot (j_{\max}(\Delta_{\min}) + 1)\cdot \Delta_{\max}$$

$$+ m\cdot (j_{\max}(\Delta_{\min}) + 2)\cdot \frac{\pi^2}{6}\cdot \Delta_{\max}$$

$$= \sum_{i\in[m]} 78\ln T \left( \frac{\Delta_{\min}^i}{f^{-1}(\Delta_{\min}^i)^2} + \int_{\Delta_{\min}^i}^{\Delta_{\max}^i} \frac{1}{f^{-1}(x)^2}\, \mathrm{d}x \right)$$

$$+ m\cdot \left[ \left( \frac{\pi^2}{6} + 1 \right) \lceil -\log_2 f^{-1}(\Delta_{\min})\rceil_0 + \frac{\pi^2}{3} + 1 \right]\cdot \Delta_{\max}.$$

(2) Take $M = a\sqrt{156m\ln T/T}$, by Lemma 11(2) and Claim 2, we continue Inequality (45) as below.

$$(45) \leq 25a\sqrt{mT\ln T} + m\cdot (j_{\max}(a\sqrt{156m\ln T/T}) + 1)\cdot \Delta_{\max}$$

$$+ m\cdot (j_{\max}(a\sqrt{156m\ln T/T}) + 2)\cdot \frac{\pi^2}{6}\cdot \Delta_{\max}$$

$$= 25a\sqrt{mT\ln T} + m\cdot \left[ \left( \frac{\pi^2}{6} + 1 \right) \lceil -\log_2(\sqrt{156m\ln T/T})\rceil_0 + \frac{\pi^2}{3} + 1 \right]\cdot \Delta_{\max}. \square$$

### F.2.2 Proof details

In this subsection, we finish the remaining part of the proof, i.e. the proofs of the claims. We first prove the bound of sufficiently sampled part, namely Claims 2. To do so, we define two kinds of niceness, that the difference between $\mu_i$ and $\hat{\mu}_i$ is small enough and that $T_i$ is large enough comparing with $N_{i,j}$, and then show that both kinds of niceness are satisfied with high probability and if so, it is impossible to play a bad action. We then prove Claim 1. In this subsection we assume $M$ is already

defined as a positive real number as in the proof of Theorem 5. Notations $\hat{\boldsymbol{\mu}}_t, \hat{\mu}_{i,t}, \bar{\boldsymbol{\mu}}_t, \bar{\mu}_{i,t}$ denote the values of $\hat{\boldsymbol{\mu}}, \hat{\mu}_i, \bar{\boldsymbol{\mu}}, \bar{\mu}_i$ at the end of round $t$, respectively.

We now prove the claims.

*Proof of Claim 2.* Explicitly,

$$
\begin{aligned}
Reg(&\{\Delta_{S_t} \geq M, \forall i, N_{i,q_i^{S_t},t-1} \geq \ell_T(\Delta_{S_t}, q_i^{S_t})\}) \\
&= \sum_{t=1}^{T} \mathbb{E}[\Delta_{S_t} \cdot \mathbb{I}\{\Delta_{S_t} \geq M, \forall i, N_{i,q_i^{S_t},t-1} \geq \ell_T(\Delta_{S_t}, q_i^{S_t})\}] \\
&\leq \sum_{t=1}^{T} \Pr\{\Delta_{S_t} \geq M, \forall i, N_{i,q_i^{S_t},t-1} \geq \ell_T(\Delta_{S_t}, q_i^{S_t})\} \cdot \Delta_{\max}.
\end{aligned}
\tag{51}
$$

We only need to bound $\Pr\{\Delta_{S_t} \geq M, \forall i, N_{i,q_i^{S_t},t-1} \geq \ell_T(\Delta_{S_t}, q_i^{S_t})\}$, i.e. the probability that for every $i$, there is $N_{i,q_i^{S_t},t-1} \geq \ell_T(\Delta_{S_t}, q_i^{S_t})$, but an action $S_t$ with $\Delta_{S_t} \geq M$ is still played. Let event $\mathcal{E}_t = \{\Delta_{S_t} \geq M, \forall i, N_{i,q_i^{S_t},t-1} \geq \ell_T(\Delta_{S_t}, q_i^{S_t})\}$. We now prove the claim that event $\mathcal{E}_t$ is not empty only when $\neg(\mathcal{N}_t^s \wedge \mathcal{N}_t^t)$, or equivalently if both the sampling and triggering are nice at the beginning of round $t$, then event $\mathcal{E}_t$ is empty. If the sampling is nice at the beginning of round $t$, then

$$
\bar{\mu}_{i,t-1} = \min\{\hat{\mu}_{i,t-1} + \rho_{i,t}, 1\} \geq \mu_i.
$$

By monotonicity, $r_S(\bar{\boldsymbol{\mu}}_{t-1}) \geq r_S(\boldsymbol{\mu})$ for every action $S$, so $\text{opt}_{\bar{\boldsymbol{\mu}}_{t-1}} \geq \text{opt}_{\boldsymbol{\mu}}$. As action $S_t$ is chosen by Oracle with input $\bar{\boldsymbol{\mu}}_{t-1}$, it must be that $r_{S_t}(\bar{\boldsymbol{\mu}}_{t-1}) = \text{opt}_{\bar{\boldsymbol{\mu}}_{t-1}} \geq \text{opt}_{\boldsymbol{\mu}}$, so $r_{S_t}(\bar{\boldsymbol{\mu}}_{t-1}) - r_{S_t}(\boldsymbol{\mu}) \geq \text{opt}_{\boldsymbol{\mu}} - r_{S_t}(\boldsymbol{\mu}) = \Delta_{S_t}$. We are going to show the claim by assuming $\mathcal{N}_t^s \wedge \mathcal{N}_t^t$ and showing $\forall i, p_i^{S_t}|\bar{\mu}_{i,t-1} - \mu_i| < f^{-1}(\Delta_{S_t})$, then by $\infty$-norm TPM bounded smoothness (Condition 6), $r_{S_t}(\bar{\boldsymbol{\mu}}_{t-1}) - r_{S_t}(\boldsymbol{\mu}) < \Delta_{S_t}$, which is a contradiction. Note that here we do need strict inequality "$<$" instead of "$\leq$" when applying Condition 6. This can be done because $i$ has at most $m$ choices and the bounded smoothness function $f$ is continuous and strictly increasing, so we can use a small enough $\varepsilon > 0$ such that $\forall i, p_i^{S_t}|\bar{\mu}_{i,t-1} - \mu_i| \leq f^{-1}(\Delta_{S_t} - \varepsilon)$, and thus $r_{S_t}(\bar{\boldsymbol{\mu}}_{t-1}) - r_{S_t}(\boldsymbol{\mu}) \leq \Delta_{S_t} - \varepsilon < \Delta_{S_t}$.

Below we omit $S_t$ from $\Delta_{S_t}, p_i^{S_t}$ and $q_i^{S_t}$. If $f^{-1}(\Delta) > p_i$, then $p_i|\bar{\mu}_{i,t-1} - \mu_i| \leq p_i|1-0| < f^{-1}(\Delta)$ without any dependency on sampling. If $f^{-1}(\Delta) \leq p_i$, then $q_i \leq 2^{\lceil -\log_2 f^{-1}(\Delta) \rceil} \leq 2^{j_{\max}(M)}$. When the sampling is nice (Definition 4), $\bar{\mu}_{i,t-1} \leq \hat{\mu}_{i,t-1} + \rho_{i,t} < \mu_i + 2\rho_{i,t}$. On the other hand, $|\bar{\mu}_{i,t-1} - \mu_i| \leq |1-0| = 1$. When the triggering is nice (Definition 7), if $\sqrt{\frac{6\ln t}{\frac{1}{3}N_{i,q_i,t-1}\cdot q_i}} \leq 1$, then $2\rho_{i,t} \leq \sqrt{\frac{6\ln t}{\frac{1}{3}N_{i,q_i,t-1}\cdot q_i}}$. So regardless whether $\sqrt{\frac{6\ln t}{\frac{1}{3}N_{i,q_i,t-1}\cdot q_i}} \leq 1$, $|\bar{\mu}_{i,t-1} - \mu_i| \leq \sqrt{\frac{6\ln t}{\frac{1}{3}N_{i,q_i,t-1}\cdot q_i}}$. Event $\mathcal{E}_t$ implies that $N_{i,q_i,t-1} \geq \ell_T(\Delta, q_i) \geq \ell_t(\Delta, q_i)$ (since $t \leq T$). So

$$
\begin{aligned}
p_i|\bar{\mu}_{i,t-1} - \mu_i| \leq p_i\sqrt{\frac{6\ln t}{\frac{1}{3}N_{i,q_i,t-1}\cdot q_i}} &\leq p_i\sqrt{\frac{6\ln t}{\frac{1}{3}\ell_t(\Delta, q_i)\cdot q_i}} < p_i\sqrt{\frac{6\ln t}{\frac{1}{3}\frac{72q_i\ln t}{f^{-1}(\Delta)^2}\cdot q_i}} \\
&= p_i\sqrt{\frac{f^{-1}(\Delta)^2}{4q_i^2}} \leq p_i\sqrt{\frac{f^{-1}(\Delta)^2}{p_i^2}} = f^{-1}(\Delta).
\end{aligned}
$$

Hence, the claim holds.

The claim implies that $\Pr\{\mathcal{E}_t\} \leq \Pr\{\neg(\mathcal{N}_t^s \wedge \mathcal{N}_t^t)\} \leq \Pr\{\neg\mathcal{N}_t^s\} + \Pr\{\neg\mathcal{N}_t^t\}$. By Lemmas 3 and 4, we have $\Pr\{\mathcal{E}\} \leq (2 + j_{\max}(M))mt^{-2}$. Plugging it into Inequality (51), we have

$$
\begin{aligned}
Reg(\{\Delta_{S_t} \geq M, \forall i, N_{i,q_i^{S_t},t-1} \geq \ell_T(\Delta_{S_t}, q_i^{S_t})\}) &\leq \sum_{t=1}^{T}(2 + j_{\max}(M))mt^{-2} \cdot \Delta_{\max} \\
&\leq m \cdot (\lceil -\log_2 f^{-1}(M) \rceil_0 + 2) \cdot \frac{\pi^2}{6} \cdot \Delta_{\max} \square
\end{aligned}
$$

*Proof of Claim 1.* Let $x$ be any real number that $x \geq M > 0$. In any round when an action $S$ with $S \in \mathcal{S}_{i,j}$ is played, $N_{i,j}$ is increased by 1. So

$$\sum_{t=1}^{T} \Pr\{S_t \in \mathcal{S}_{i,j}, N_{i,j,t-1} < \ell_T(x, 2^{-j})\} \leq \ell_T(x, 2^{-j}).$$

If we add an additional restriction $\Delta_{S^t} \geq x$, the probability will not increase, so

$$\sum_{t=1}^{T} \Pr\{\Delta_{S_t} \geq x, S_t \in \mathcal{S}_{i,j}, N_{i,j,t-1} < \ell_T(x, 2^{-j})\} \leq \ell_T(x, 2^{-j}).$$

We use the shorthand $\mathcal{E}_{i,j}^{S_t}$ to denote the event $\{S_t \in \mathcal{S}_{i,j}, N_{i,j,t-1} < \ell_T(x, 2^{-j})\}$. Suppose $X$ is a non-negative random variable with $\Pr\{X \geq M\} = p$ and $\Pr\{X = 0\} = 1 - p$. Then by the basic principal on expectation, we have

$$\mathbb{E}[X] = \int_0^{+\infty} \Pr\{X \geq x\}\, \mathrm{d}x = \int_0^M \Pr\{X \geq x\}\, \mathrm{d}x + \int_M^{+\infty} \Pr\{X \geq x\}\, \mathrm{d}x$$
$$= pM + \int_M^{+\infty} \Pr\{X \geq x\}\, \mathrm{d}x.$$

Applying the above, we have

$$Reg(\{\Delta_{S_t} \geq M\} \cap \mathcal{E}_{i,j}^{S_t})$$
$$= \sum_{t=1}^{T} \mathbb{E}[\mathbb{I}(\{\Delta_{S_t} \geq M\} \cap \mathcal{E}_{i,j}^{S_t}) \cdot \Delta_{S_t}]$$
$$= \sum_{t=1}^{T} \left(\Pr[\{\Delta_{S_t} \geq M\} \cap \mathcal{E}_{i,j}^{S_t}] \cdot M + \int_M^{+\infty} \Pr[\{\Delta_{S_t} \geq x\} \cap \mathcal{E}_{i,j}^{S_t}]\, \mathrm{d}x\right)$$
$$= \sum_{t=1}^{T} \Pr[\{\Delta_{S_t} \geq M\} \cap \mathcal{E}_{i,j}^{S_t}] \cdot M + \int_M^{+\infty} \sum_{t=1}^{T} \Pr[\{\Delta_{S_t} \geq x\} \cap \mathcal{E}_{i,j}^{S_t}]\, \mathrm{d}x$$
$$= \sum_{t=1}^{T} \Pr[\{\Delta_{S_t} \geq M\} \cap \mathcal{E}_{i,j}^{S_t}] \cdot M + \int_M^{\max\{\Delta_{\max}^i, M\}} \sum_{t=1}^{T} \Pr[\{\Delta_{S_t} \geq x\} \cap \mathcal{E}_{i,j}^{S_t}]\, \mathrm{d}x$$
$$\leq \ell_T(M, 2^{-j})M + \int_M^{\max\{\Delta_{\max}^i, M\}} \ell_T(x, 2^{-j})\, \mathrm{d}x. \qquad \square$$

### F.3 Comparison between 1-norm and $\infty$-norm

In this paper, we give upper bounds of regret for CMAB-T problems that satisfy TPM bounded smoothness with 1-norm or with $\infty$-norm. We emphasis Theorem 1 and Theorem 5 do not imply each other. For clarity, we use $a_1$ and $a_\infty$ in place of $a$ in bounded smoothness function $f(x) = ax$. If a CMAB-T problem instance satisfies TPM bounded smoothness with 1-norm with $f(x) = a_1 x$, then it also satisfies TPM bounded smoothness with $\infty$-norm with $f(x) = a_\infty x$, where $a_\infty = K a_1$. Conversely, if a CMAB-T problem instance satisfies TPM bounded smoothness with $\infty$-norm with $f(x) = a_\infty x$, then it also satisfies TPM bounded smoothness with 1-norm with $f(x) = a_1 x$, where $a_1 = a_\infty$. For distribution-dependent upper bound, according to Theorems 1 and 5, we have $O(\frac{a_\infty^2 m \ln T}{\Delta})$ and $O(\frac{a_1^2 K m \ln T}{\Delta})$. For a problem instance that satisfies TPM bounded smoothness with 1-norm with $f(x) = a_1 x$, if we use the bound for $\infty$-norm, the result will be $O(\frac{a_1^2 K^2 m \ln T}{\Delta})$. For a problem instance that satisfies TPM bounded smoothness with $\infty$-norm with $f(x) = a_\infty x$, if we use the bound for 1-norm, the result will be $O(\frac{a_\infty^2 K m \ln T}{\Delta})$. Both give an additional $K$ factor. It is similar for distribution-independent bound, which will have an additional $\sqrt{K}$ factor in both cases.