[Reviews · NeurIPS 2017]

Reviewer 1



The paper studies the stochastic combinatorial semi-bandit problem. The authors use a general framework in which the actions trigger the arms in a stochastic fashion. Previous studies have that the minimal probability that an arm is triggered appears in the upper bounds proved on the expected regret of the state-of-the art-algorithm. This paper shows that under some smoothness condition this annoying term will not appear in the bound. Intuitively, if the contribution of the arm to the expected rewards of any solutions is small then we don't need much information about it. Moreover, they prove that in general the minimal probability term is unavoidable by providing a lower bound that contains this term. The paper is clearly written. The contribution of the paper is mostly technical but is of interest. Providing a lower bound and a nice condition in which case the minimal probability term disappears is nice though still somehow unsurprising. The analysis of the paper applies to a large general framework and make a flurry of contributions to a series of related works. Can you comment of the assumption in line 182. How big is this assumption? It seems to me that it is a pretty constraining assumption that limits the reach of the paper. Is there a lot of interesting examples where the probability of triggering base arm i with action S is determined by the expectation vector μ ? Line 54-55: weird formatting.

Reviewer 2



The authors condider the problem of Combinatorial Semi-Bandits with triggered arms. By introducing a new assumption based on triggered probability, they can get rid of a factor 1/p in the regret bounds, where p* is the smallest triggered probability. The literature is well-cited and the paper is well-written. I really enjoyed how there general general analysis allows to recover and improve several existing settings: CMAB, influence maximization, cascading bandits or linear bandits. There assumption is furthermore quite intuitive: the less an arm can be triggered, the lower is his impact on the expected reward. I however did not have time to read the proofs in the appendix. 1. Wouldn't the analysis could be performed for arbitrary unknown non-parametric set of distribution D, since the algorithm only uses the mean estimates. 2. Your lower bound uses rewards in [0,1/p]. What about the case were the rewards are bounded in [0,1]? Do you think, the factor 1/p is still necessary? 3. Do you have examples, where the set of super-arms is infinite or continuous while there are only a finite number of arms? Typos: - l55: "both" -> "Both" - l236: "[1,1]": is it necessary? - l237: I did not understand why a set of super-arms j, only generate regret at most "O(2^{-j} log T/\delta_i^2)"? Isn't it the regret of a single super-arm in this set?

Reviewer 3



Overview: The paper studies combinatorial multi-armed bandit with probabilistically triggered arms and semi-bandit feedback (CMAB-T). The paper improves the existing regret bounds by introducing a new version of the bounded smoothness condition on the reward functions, called triggering probability modulated bounded smoothness condition. The intuition behind this is that hard-to-trigger arms do not contribute much to the expected regret. With this new condition, the authors are able to remove a potentially exponentially large factor in the existing regret bounds. The paper then shows that applications such as influence maximization bandit and combinatorial cascading bandit both satisfy this condition. Finally, the paper gives a lower bound and shows that for general CMAB-T problems, the exponential factor is unavoidable. Quality: The paper provides rigorous proofs for all its claims. Clarity: The paper is well-organized and clearly-written. Everything is well-defined and well-explained. The paper also has a nice flow to it. My only complaint is that a concluding section is missing. The authors may also want to proofread for typos and grammatical errors. Originality: The paper strengthens an existing condition and gives a tighter regret bound for CMAB-T problems. The paper also applies this new condition to influence maximization bandit and combinatorial cascading bandit. The authors introduce several new techniques in their proofs. Significance: The paper greatly improves the regret bound for CMAB-T problems with a new condition that has many applications.